

# Can large-scale tree cover change negate climate change impacts on future water availability?

Freek Engel[1], Anne J. Hoek van Dijke[2], Caspar T.J. Roebroek[3, 4, 5], and Imme Benedict[1]

[1]Meteorology and Air Quality Group, Wageningen University and Research, Wageningen, The Netherlands
[2]Max Planck Institute for biogeochemistry, Jena, Germany
[3]European Commission, Joint Research Centre (JRC), Ispra, Italy
[4]Hydrology and Water Management group, Wageningen University and Research, Wageningen, The Netherlands
[5]Institute for Atmospheric and Climate Science, Eidgenössische Technische Hochschule (ETH) Zurich, Zurich, Switzerland

**Correspondence:** Freek Engel (freek.engel@wur.nl) and Imme Benedict (imme.benedict@wur.nl)

**Abstract.** The availability of fresh water over land may become increasingly scarce under climate change, and natural and human-induced tree cover changes can further enhance or negate the water scarcity. Previous studies showed that global tree cover change can have large impacts on water availability under current climate conditions, but did not touch upon the implications of global tree cover change under climate change. Here, we study the hydrological impacts of large-scale tree cover

change (climate-induced changes in combination with large-scale afforestation) in a future climate (SSP3-7.0) following an interdisciplinary approach. By combining data from five CMIP6 climate models with a future potential tree cover dataset, six Budyko models, and the UTrack moisture recycling dataset, we can disentangle the impacts of climate change and future tree cover change on evaporation, precipitation, and runoff. We quantify per grid cell and for five selected river basins (Yukon, Mississippi, Amazon, Danube, and Murray-Darling) if tree cover changes enhance or counteract the climate-driven changes in

runoff due to their impact on evapotranspiration and moisture recycling. Globally averaged, the impacts of climate change and large-scale tree cover change on runoff are of similar magnitude with opposite signs. While climate change increases the global runoff, the changing tree cover reverses this effect which overall results in a limited net impact on runoff relative to the present climate and current tree cover. Nevertheless, locally the change in runoff due to tree cover change and climate change can be substantial with increases and decreases of more than $100\,\mathrm{mm\,yr^{-1}}$. We show that for approximately $16\,\%$ of the land surface,

tree cover change can increase the water availability significantly. However, we also find that, for $14\,\%$ of the land surface, both tree cover change and climate change might decrease water availability with more than $5\,\mathrm{mm\,yr^{-1}}$. For each of the selected catchments, the direction and magnitude of the impacts of climate change and tree cover change vary, with dominating climate change impacts in all basins except the Mississippi River basin. Our results show that ecosystem restoration projects targeting an altered tree cover should consider the corresponding hydrological impacts to limit unwanted (non-)local reductions in water

availability.



## 1 Introduction

Forests play an important role in, among others, the conservation of biodiversity, reduction of soil erosion, and mitigation of climate change (Herrick et al., 2019). Therefore, there have been many local and global initiatives to increase tree cover, such as the Bonn challenge, Grain for Green, the 20x20 Initiative, Billion Trees Campaign, and AFR100. Climate change, especially
climatic drying, has consequences for the ability to restore forests. At the same time, large-scale forest restoration has major impacts for local and global water availability, and can reduce water availability in water scarce regions (Hoek van Dijke et al., 2022).

Over the past decades, both climate warming and land cover changes have impacted global freshwater availability by changing evaporation and precipitation. While recent climate warming likely enhanced global evaporation and precipitation over
land (Douville et al., 2021), the simultaneous changes in land cover showed contrasting impacts on evaporation. For example, land-cover changes that occurred between 1950 – 2000 decreased evaporation by $5\%$ (Sterling et al., 2013), whereas from the 1980s onwards, global vegetation greening increased evaporation with $3.7\%$ (Yang et al., 2023). Teuling et al. (2019) showed that both climate warming and land cover changes impacted evaporation over Europe with large regional differences. For instance, while evaporation increased with more than $15\%$ between 1960 – 2010, it decreased for parts of Southern Europe
(Teuling et al., 2019). Furthermore, the greening of vegetation in China enhanced the yearly precipitation due to an increase in convective precipitation (Yu et al., 2020). These changes in evaporation and precipitation have impacted streamflow globally (Sterling et al., 2013; Teuling et al., 2019; Piao et al., 2007; Wang-Erlandsson et al., 2018), and thereby affected the occurrence of floods and droughts, production of hydropower, and availability of drinking and irrigation water. However, the projected increases in climate warming and land cover changes in the (near) future could impact the water fluxes and availability even
further.

Since climate warming can increase the energy available for evaporation and enhances the moisture holding capacity of air, these effects are expected to further increase evaporation, mean precipitation, and extreme precipitation events (Trenberth, 2011). At the same time, the future holds longer periods of dry spells and droughts in many regions (Milly and Dunne, 2016), and therefore water scarcity will likely increase for a growing fraction of land, impacting many lives and ecosystems (Caretta
et al., 2022).

The combination of climate warming and human interventions, such as afforestation and deforestation, can affect the global tree cover in a future climate. Roebroek (2023) recently calculated a future potential tree cover, representing the maximum number of trees that could grow on Earth given the climate and soil characteristics, for different climate pathways. The study found that a warmer climate can support tree growth in the colder high latitudes, whereas increasing aridity is expected to
reduce the forest cover in the American and African tropics (see Fig. 1, showing the differences between the present and future potential tree covers). At the same time, there is a strong global incentive to increase tree cover due to the benefits for climate mitigation, ecosystem restoration, and conservation of species (Bastin et al., 2019; Griscom et al., 2017). Roebroek (2023) showed that global climate change decreases the overall global potential for tree restoration, however, the tree carrying



capacity is 55 % above the current tree cover. This estimation of potential tree cover change under climate change allows us to
study the combined effects of future global warming and tree cover change on water availability.

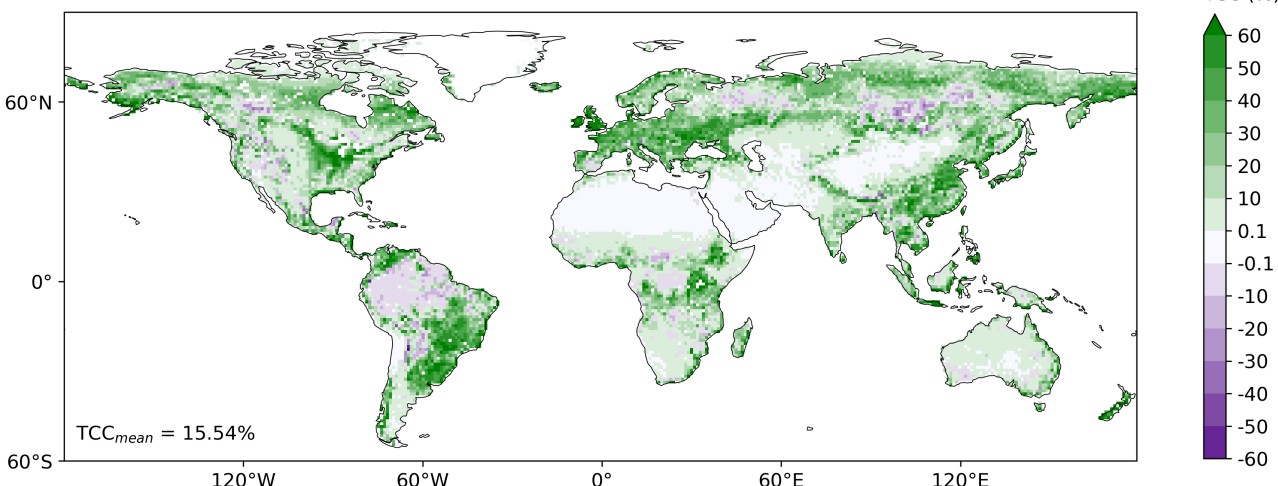

**Figure 1.** Tree cover change (TCC) in percentages, the TCC is defined as the difference between the potential tree cover in a future climate (2041-2060) and the tree cover in a present climate (2000). For the future potential tree cover we use the average potential tree cover data based on bioclimatic variables, derived from the outputs of five CMIP6 models (CMCC-ESM2; INM-CM5-0; IPSL-CM6A-LR; MIROC6; and UKESM1-0-LL) under climate change pathway SSP3-7.0.

Changes in tree cover impact local and regional water availability through their effect on evapotranspiration and precipitation (Ellison et al., 2017; Hoek van Dijke et al., 2022; King et al., 2024; Zhang et al., 2022). Forest evapotranspiration is higher compared to evapotranspiration from short vegetation and bare land (Zhang et al., 2001) since trees increase the availability of energy (through albedo), and water (through their deep rooting systems) for evapotranspiration, and increase the aerodynamic
conductance. In addition, trees can impact precipitation directly by locally increasing convection and turbulence in the atmosphere, while trees can also affect precipitation indirectly by enhancing the recycling of evaporated moisture both locally or regionally (Meier et al., 2021; Ellison et al., 2017; De Hertog et al., 2023; Wang-Erlandsson et al., 2018). Therefore, future forest restoration projects can potentially be used to increase water availability in water scarce regions (Staal et al., 2024b).

Previous literature has already explored the effects of climate change and large-scale tree cover change, nevertheless, un-
certainty persists regarding the separate and combined impacts on global water availability in a future climate. Tuinenburg et al. (2022) analysed how the current potential tree cover on a global scale could mitigate future drying trends, however, this study only focused on global precipitation and did not consider an altered potential tree cover in a future climate. King et al. (2024) applied a global forestation scenario in one earth system model and found regional reductions in water availability up to 15%. More local studies showed that in mountainous catchment areas, climate-driven changes in vegetation could mitigate or
reverse the climate-driven increases in runoff (Rasouli et al., 2019). Another example is that in Northern China the soil water





availability decreased before 2000 and increased afterwards due to climate change and vegetation greening, although the effect of vegetation greening was minor compared to the climate change effect (Douville et al., 2021). Given the remaining uncertainties, a global comprehensive analysis regarding the impacts of tree cover and climate change, and their separate contributions, is needed which can help to secure freshwater availability in the future, and to support strategic tree restoration planning.


Here, we study the combined and separate impacts of climate change and large-scale tree cover change on terrestrial evaporation, precipitation, and runoff worldwide. We take a data-driven interdisciplinary approach, by combining tree cover maps with Budyko models and projected climate data from the Coupled Model Intercomparison Project phase 6 (CMIP6), to calculate present and future water fluxes. The future climate data is obtained for the Shared Socioeconomic Pathway (SSP) with

an intermediate to strong climate change signal (SSP3-7.0). This climate pathway involves high greenhouse gas emissions which result in an average increase in global surface air temperature of $1.4°$ for 2041–2060 relative to 1995–2014 (Lee et al., 2021). Our implemented tree cover change scenario represents the tree cover carrying capacity under climate pathway SSP3-7.0 and includes natural changes in tree cover, as well as, human-induced large-scale tree planting. In this study we compare the impacts of climate change and tree cover change on water fluxes over land and analyse where these effects can enhance or

counteract each other on grid cell level and at catchment scale. For five selected catchments we provide insight whether climate or large-scale tree cover change can be a dominant driver of water availability change in a future climate.

## 2    Methodology

There are three research scenarios that guide this work to investigate the impact of climate change and tree cover change on the hydrological fluxes precipitation (P), evapotranspiration (ET), and runoff (Q, also referred to as water availability). Those three

scenarios are: 1) scenario present climate, 2) scenario climate change (CC), and 3) scenario climate change with tree cover change and moisture recycling change (CC+TCC) (Table 1). All research scenarios use P and potential evapotranspiration (PET) datasets from five CMIP6 climate models (Sect. 2.1), a tree cover dataset (Sect. 2.2), and Budyko models (Sect. 2.3) for the calculation of ET and Q fluxes. Furthermore, scenario CC + TCC includes the (non-)local indirect effects of changes in ET and P by accounting for an altered moisture recycling, obtained with the moisture tracking dataset UTrack (Sect. 2.4). All

datasets in this study are reprojected to a spatial resolution of $1°$ by $1°$ to ensure compatibility with the spatial resolution of the UTrack dataset. In addition, datasets with a time dimension are aggregated to yearly averaged data for the use of the Budyko models. An overview of the research scenarios with the corresponding input and output datasets and calculations is shown in Table 1. The analyses in this study are performed at a global scale and for five river basins: Yukon River basin, Mississippi River basin, Amazon River basin, Danube River basin, and the Murray-Darling River basin.

### 2.1   Climate input data from CMIP6 climate models

Five global climate model simulations from CMIP6 are selected to provide P and PET datasets for the different research scenarios in this study. For the present climate scenario, climate data is obtained from historical CMIP6 simulations for the



**Table 1.** Overview of the research methodology and input and output data for each of the research steps. The abbreviations represent; P: precipitation from CMIP6, PET: potential evapotranspiration from CMIP6, ET: evapotranspiration, Q: runoff. Furthermore, 'mean' refers to the mean flux over the CMIP6-Budyko models and 'std' refers to the standard deviation of the flux over the CMIP6-Budyko models. Note that five CMIP6 models and six Budyko models result in 30 CMIP6-Budyko combinations for ET and Q. The general research methodology is adopted from Hoek van Dijke et al. (2022). For more information about the datasets and their sources, Table A1 provides an overview of the datasets used within our study.

| Research scenario | Tree cover | Climate pathway | Calculations with six Budyko models | | UTrack | |
| | | | Climate input data | Budyko output data | Input | Output |
|---|---|---|---|---|---|---|
| 1. Present climate | Present tree cover (2000) | Historical (1985–2014) | 5x P, 5x PET | 30x ET, $\to ET_{mean}, ET_{std}$ <br> 30x Q $\to Q_{mean}, Q_{std}$ | | |
| 2. Climate change (CC) | | SSP3-7.0 (2035–2064) | 5x P, 5x PET | 30x ET, ↘ <br> 30x Q 30x $\Delta ET \to \Delta ET_{mean}$ | $\Delta ET_{mean}$ | $\Delta \mathbf{P_{mean}}$ |
| Climate change with tree cover change (intermediate step) | Future potential tree cover (2041–2060) | | | 30x ET, ↗ <br> 30x Q | | |
| 3. Climate change with tree cover change with changed moisture recycling (CC+TCC) | | | 5x P+ $\Delta \mathbf{P_{mean}}$, 5x PET | 30x ET, $\to ET_{mean}, ET_{std}$ <br> 30x Q $\to Q_{mean}, Q_{std}$ | | |

time interval 1985–2014 and for the future climate scenarios, data is used from ScenarioMIP simulations under climate change pathway SSP3-7.0 for 2035–2064 (Table 1) (O'Neill et al., 2016). The P datasets are retrieved directly from the CMIP6 models,
however, due to the limited availability of global PET output from CMIP6 models for climate pathway SSP3-7.0, we use the PET datasets provided by Bjarke et al. (2023). Bjarke et al. (2023) calculated PET with the Priestley-Taylor method (Priestley and Taylor, 1972) based on the sensible and latent heat fluxes, mean surface air temperature, and surface air pressure for which data was directly retrieved from selected CMIP6 climate models for both the historical and ScenarioMIP simulations.

The five CMIP6 models in this study are selected based on the following criteria; 1) the model should have PET output
provided by Bjarke et al. (2023) and a future tree cover dataset for ScenarioMIP SSP3-7.0 from Roebroek (2023) (Sect. 2.2); 2) the spatial resolution of the CMIP6 model data should be close to the $1°$ by $1°$ spatial resolution used in this study; 3) the absolute percentual bias of the calculated historical PET compared to the ERA5Land dataset, as provided in Table 2 from Bjarke et al. (2023), should be smaller than $10\%$; and 4) from each institute only one CMIP6 model is selected. The aforementioned criteria result in the selection of the following five CMIP6 models: CMCC-ESM2; INM-CM5-0; IPSL-CM6A-LR; MIROC6;
and UKESM1-0-LL, for which an overview is provided in Table 2.



**Table 2.** Overview of the models from phase 6 of the Coupled Model Intercomparison Project (CMIP6) from which climate data for precipitation (P) and potential evapotranspiration (PET) was used in this study. The climate data is obtained for variant label 'r1i1p1f1' for all models, except for UKESM1-0-LL for which 'r1i1p1f2' is used due to data availability restrictions, meaning that a different forcing dataset was used for this model simulation in UKESM1. The lon x lat resolution in the table refers to the number of gridcells present for the longitude and latitude dimensions within the global dataset.

| CMIP6 model | Institution (institution ID) | Original resolution (lon x lat) for P and PET | Model type* | Reference |
|---|---|---|---|---|
| CMCC-ESM2 | Fondazione Centro Euro-Mediterraneo (CMCC) | 288 x 192 | Earth system | Lovato et al. (2022) |
| INM-CM5-0 | Institute for Numerical Mathematics (INM) | 180 x 120 | Atmosphere-ocean general circulation | Volodin et al. (2017) |
| IPSL-CM6A-LR | Institut Pierre Simon Laplace (IPSL) | 144 x 143 | Earth system | Boucher et al. (2020) |
| MIROC6 | Atmosphere and Ocean Research Institute, The University of Tokyo and Japan Agency for Marine-Earth Science and Technology (MIROC) | 256 x 128 | Atmosphere-ocean general circulation | Tatebe et al. (2019) |
| UKESM1-0-LL | Met Office Hadley Centre (MOHC) | 192 x 144 | Earth system | Sellar et al. (2019) |

* As classified by Kuma et al. (2023)

The monthly P ($\mathrm{kg\,m^{-2}\,s^{-1}}$) and PET ($\mathrm{mm\,d^{-1}}$) datasets are aggregated to yearly data and averaged over a 30 year period ($\mathrm{mm\,yr^{-1}}$) as indicated in Table 1. These yearly mean P and PET datasets are used as input for the Budyko calculations (Table 1 and Sect. 2.3). Note that large water bodies, the Antarctic region, and most of Greenland are masked out.

To gain insight into the differences between the five climate models, Fig. 2a shows PET and P over land in relation to the aridity index (PET / P) for a future climate. The variability between the models is highest for terrestrial P over humid regions, ranging from $837\,\mathrm{mm\,yr^{-1}}$ in the INM model to $2782\,\mathrm{mm\,yr^{-1}}$ in the CMCC model (Fig. 2a). The high variability in P between CMIP6 models over wet regions is also underlined by Yazdandoost et al. (2021) for historical data. The model spread for humid regions is also present for PET, however, the largest variability for PET is found over arid regions.

**2.2    Tree cover datasets**

In this study, two tree cover datasets are applied: 1) a present tree cover dataset used for scenario present climate and scenario CC, and 2) a future potential tree cover dataset, representing the maximum tree carrying capacity under the studied climate pathway, which is used for scenario CC+TCC (Table 1). For the present tree cover, we use the global tree cover dataset for the year 2000 provided by Hansen et al. (2013), which is generated with Landsat satellite data. For the future potential tree cover



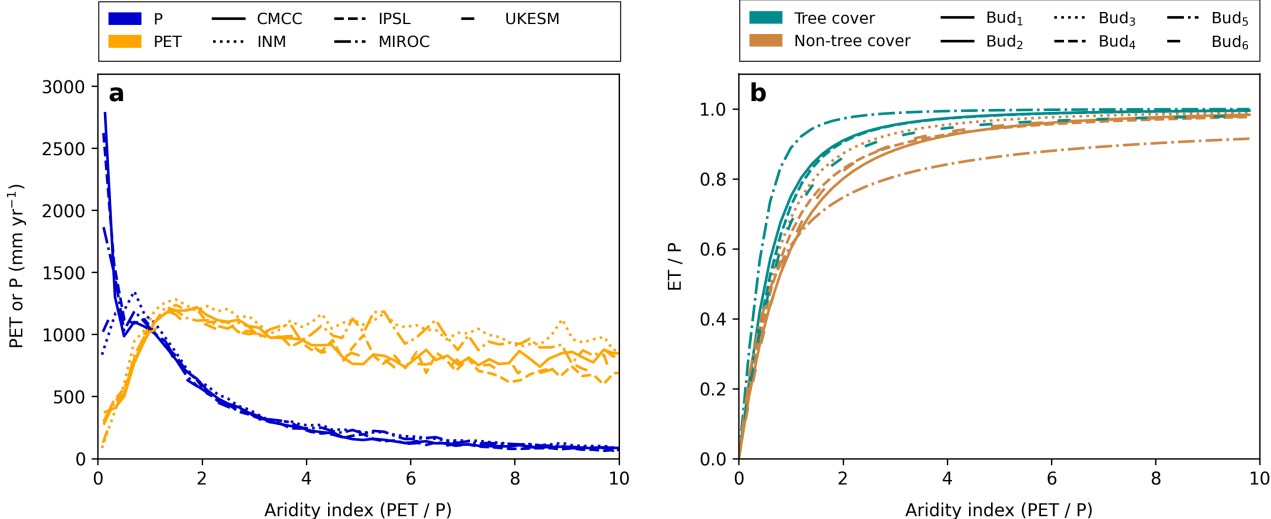

**Figure 2.** Illustration of variability over the CMIP6 and Budyko models in relation to the aridity index (PET / P); a) variability in terrestrial precipitation (P) and potential evapotranspiration (PET) fluxes for five CMIP6 models (CMCC-ESM2; INM-CM5-0; IPSL-CM6A-LR; MIROC6; and UKESM1-0-LL) under climate change pathway SSP3-7.0. b) variability in the evapotranspiration over precipitation ratio (ET / P) for tree cover and non-tree cover for six Budyko (Bud) models, the numbering of these models is consistent with the numbering in Table A2.

dataset, we use a dataset by Roebroek (2023) that predicts the tree cover carrying capacity for different climate scenarios, and is adjusted for the natural occurrence of disturbances (e.g., forest fires, wind throws, insect outbreaks etc.). This tree cover data, here referred to as 'future potential tree cover' for simplicity, is created by integrating the previously mentioned tree cover (Hansen et al., 2013) and climate characterisations from the WorldClim V2 dataset (Fick and Hijmans, 2017) in a machine learning framework. Note that the dataset represents the natural capacity of the earth to support trees, and thus would allow
for tree restoration on agricultural or urban land. In this study, we retrieved the future potential tree cover datasets for the time range 2041–2060 under climate pathway SSP3-7.0 for each of the five selected CMIP6 models (Sect. 2.1) to create an averaged future potential tree cover dataset for scenario CC+TCC. The future potential tree cover maps represent a shorter time period (20 years) than the climatological data (30 years), as the future potential tree cover maps are only available for twenty-year periods (Roebroek, 2023). We assume that the tree cover datasets for both the present and the future time period consist of
mature trees.

The tree cover change illustrates the differences between the two tree cover datasets, which is calculated by subtracting the present tree cover from the future potential tree cover (Fig. 1). Note that the tree cover change is not used for calculations and hence is solely used to visualize the differences in tree cover between the scenarios present climate and CC, and scenario CC+TCC. Globally, there is an average increase in tree cover of 15.5 %. Decreases in tree cover are found over the Amazon,
in northwestern North America, and in middle north Eurasia.



## 2.3 Budyko model calculation

For each research scenario, yearly mean ET and Q are calculated using the Budyko framework (Budyko, 1974), following the approach used by Hoek van Dijke et al. (2022); Teuling et al. (2019). A Budyko model describes the long-term mean partitioning of P into ET and Q, as a function of aridity (Fig. 2b) and has a general form like:

$$\frac{ET}{P} = f(\frac{PET}{P}, \omega) \tag{1}$$


where $\frac{ET}{P}$ is the fraction of precipitation partitioned into evaporation, $\frac{PET}{P}$ is the aridity index and $\omega$ is a model parameter. Previous studies have shown that $\omega$ is closely related to vegetation type (e.g. Zhang et al., 2001). In our study, we use an ensemble of six different Budyko models, that have a different model formulation, but represent a similar curve (Fig. 2, model equations in Table A2; Zhang et al. (2001, 2004); Zhou et al. (2015); Teuling et al. (2019); Oudin et al. (2008)). After

calculating the $\frac{ET}{P}$ fraction with a Budyko model, the $\frac{Q}{P}$ fraction is obtained by rewriting a simplified water balance for a multi-year timescale (Q = P-ET) to $\frac{Q}{P} = 1-\frac{ET}{P}$. In this study, ET and Q are calculated by using P and PET datasets from the five CMIP6 models in combination with the fractional tree cover from the present and the future potential tree cover datasets following Table 1. For each grid cell, values for ET and Q are calculated as a fraction of tree-covered and non-tree-covered surfaces relative to their occurrence in the grid cell.

Fig. 2b shows the variability between the six Budyko models for a theoretical ET/P fraction in relation to a theoretical aridity index (PET/P) for (non-)tree-covered surfaces. The figure shows that there can be pronounced variability between the different Budyko models, however, it is also shown that for all models a larger fraction of P is evaporated for tree-covered surfaces compared to non-tree-covered surfaces. The combined variability of the climate input data from the CMIP6 models (Fig. 2a) and the variability between the Budyko models (Fig. 2b) provides an uncertainty estimate for the calculated ET and Q in this

study.

## 2.4 UTrack moisture tracking dataset

Following large-scale tree cover change, the increase in terrestrial ET will increase the (terrestrial) P both locally and remotely through moisture recycling. In scenario CC+TCC we account for this local and remote change in P by using the UTrack dataset, created by Tuinenburg et al. (2020), in which the moisture recycling is quantified per grid cell. This dataset is generated by

combining ERA5 reanalysis data with UTrack, a Lagrangian model that tracks the transport of moisture through the atmosphere (Tuinenburg and Staal, 2020). The UTrack dataset contains atmospheric moisture trajectories, averaged over 2008-2017 (Tuinenburg et al., 2020), that show the transport of evaporated moisture from a source location to precipitated moisture at a target location. We retrieved the dataset at a spatial resolution of $1°$ by $1°$ for each month of the year and we aggregated these monthly moisture trajectories to the yearly timescale. The yearly average moisture trajectories were used to calculate how

the mean tree cover-driven change in ET affects the global P flux (Table 1). The use of yearly trajectories was preferred over monthly trajectories since the Budyko approach is only valid on multi-year timescale. Hoek van Dijke et al. (2022) showed





that temporal aggregation of UTrack, rather than temporal disaggregation of Budyko, resulted in similar patterns for moisture recycling. It should be noted that altering the tree cover creates a feedback loop where a change in terrestrial ET can in turn affect the (terrestrial) P, which again impacts ET, although the impact on both fluxes becomes increasingly smaller for each
cycle. In this study, the changed moisture recycling is calculated twice for scenario CC+TCC, and the impacts of the changed moisture recycling on the P, ET, and Q fluxes are relatively small after the UTrack dataset is applied the second time (Table A3 and Table A4).

We underline that the UTrack dataset represents current atmospheric conditions (Tuinenburg et al., 2020) and therefore, this
study does not account for possible changes in moisture recycling due to changes in circulation which result from climate change. The UTrack dataset also does not include the feedbacks of an altered tree cover, even though tree cover change can impact the atmospheric conditions and circulation (De Hertog et al., 2023; Portmann et al., 2022; Davin and de Noblet-Ducoudré, 2010; Duveiller et al., 2018). Nevertheless, most moisture tracking models that are currently available rely on meteorological reanalysis data and thus these models are only valid under current climate and land-cover conditions. Hence,
given the unavailability of a moisture tracking dataset for a future climate, and the approach of combining climate datasets instead of actively running a global climate model, this study relies on the UTrack dataset for the future moisture recycling. The implications of this approach are further discussed in Sect. 4.

## 3   Results and Discussion

### 3.1   Global mean impact of climate change and future tree cover change on hydrological fluxes

First, we compare our global mean hydrological fluxes over land for the present climate with multi-model outcomes from CMIP6, after which we discuss the impacts of scenarios climate change (CC) and climate change with tree cover change (CC+TCC). For the present climate scenario, the mean fluxes in this study are $565\,\mathrm{mm\,yr^{-1}}$ for ET over land and $386\,\mathrm{mm\,yr^{-1}}$ for Q (Fig. 3a). Our mean value of ET exceeds the $90\,\%$ confidence range for the multi-model averages over land by IPCC of $482-544\,\mathrm{mm\,yr^{-1}}$, obtained with 32 CMIP6 models for $1995-2014$, whereas our mean Q value is consistent with its
corresponding range of $179-460\,\mathrm{mm\,yr^{-1}}$ (Douville et al., 2021). Note that our mean terrestrial P flux of $952\,\mathrm{mm\,yr^{-1}}$ also slightly exceeds the multi-model range of $723-942\,\mathrm{mm\,yr^{-1}}$ by Douville et al. (2021). This indicates that the hydrological fluxes for our present climate scenario ($1985-2014$) for the five selected CMIP6 models in this study are at the higher end of the multi-model range.

Under climate change (scenario CC), the terrestrial P increases with $33\pm55\,\mathrm{mm\,yr^{-1}}$ and terrestrial ET increases with
$22\pm24\,\mathrm{mm\,yr^{-1}}$, resulting in an average increase for Q of $11\pm39\,\mathrm{mm\,yr^{-1}}$ (Fig. 3b, Table A3). These flux differences relative to the present climate were calculated for each of the 30 CMIP6(-Budyko) model combinations, generating 30 $\Delta$flux values for each land grid cell after which the mean and standard deviation were computed for each grid cell. Although the mean changes for the hydrological fluxes are consistent with the climate change impacts described by Douville et al. (2021), in which 32 CMIP6 models are compared, our study shows larger corresponding standard deviations. These high standard deviations arise



from deviations in both the climate input data as well as the use of six different Budyko models (Fig. 2). Especially the Q flux shows a relatively large variability compared to the other hydrological fluxes, which is also mentioned by Li and Li (2022) for SSP1-2.6, SSP2-4.5, and SSP5-8.5.

Implementing the future potential tree cover in a future climate (scenario CC+TCC) generates mean hydrological impacts that are of similar magnitude as the effects of climate change (scenario CC). The changing tree cover enhances mean global ET with $28 \pm 19 \, \mathrm{mm \, yr^{-1}}$ ($4.7\,\%$) relative to scenario CC due to an average global tree cover increase of $15.5\,\%$. This increase in ET includes both the direct and indirect tree cover change effects, of which the latter relates to the altered P flux. The enhanced ET can affect P over both land and ocean surfaces (Fig. A1), enhancing P with $16 \, \mathrm{mm \, yr^{-1}}$ over land. Please note that the standard deviation for the altered P remains unchanged across scenarios CC and CC+TCC since this standard deviation only shows the (constant) variability over the CMIP6 models. We find that approximately $60\,\%$ of the additional evaporated moisture precipitates over land. The climate-driven net increase in Q is shifted to a slight net decrease after tree cover change (scenario CC+TCC). Hence, globally, the contrasting effects of climate change and tree cover change generate a limited net effect for Q relative to the present climate (Fig. 3b). A similarity in magnitude of impact for future vegetation change and climate change on Q is also demonstrated regionally by Rasouli et al. (2019) for mountainous catchments in North America, although the impact extents can vary among catchments (Rasouli et al., 2019). However, in our study, the tree cover-driven impacts on the Q flux can differ substantially on a regional level and can therefore deviate from the global effect, as further discussed in Sect. 3.2 and Sect. 3.3.

## 3.2 Spatial impact of climate change and future tree cover change on hydrological fluxes

Under climate change, there is generally an increase in P and ET fluxes over land, however, there are also regions in e.g., South America, southern Europe, Africa and Australia where decreases in both P and ET are found (Fig. 4a, 4b). P and ET can experience pronounced climate-driven absolute changes in the tropical belt, although it should be noted that the magnitude of relative changes can be comparatively smaller in these regions (not shown). These spatial trends for P and ET over land agree with the multi-model ensemble projections for climate pathways SSP2-4.5 and SSP5-8.5 (Zhao and Dai, 2021; Li and Li, 2022). When including the changes in tree cover, the patterns for tree cover change (Fig. 1) are mirrored in the tree cover-driven changes in ET (Fig. 4f); hence an increase in tree cover enhances ET and a decrease in tree cover reduces ET. This causal relationship for large-scale tree cover increase and ET is also shown under present climate conditions (Tuinenburg et al., 2022; De Hertog et al., 2023). The change in ET per change in tree cover is largest in wet regions, which is why we find a large increase in ET over Southeast Asia, tropical Africa and eastern South America. Since only a small part of the recycled moisture generally precipitates within $100 \, \mathrm{km}$ of the evaporation source (Cui et al., 2022; Theeuwen et al., 2023), a change in local ET will mainly affect the downwind P at regional (or larger) spatial scales. Hence, the tree cover change patterns are less distinguishable for the change in P which is, for example, illustrated with the absence of negative P change over land despite the negative tree cover changes (Fig. 4e). This absence of negative P could be attributed to a compensation effect where small negative local P changes are compensated by increased P originating from other (upwind) areas, as also indicated by Tuinenburg et al. (2022).





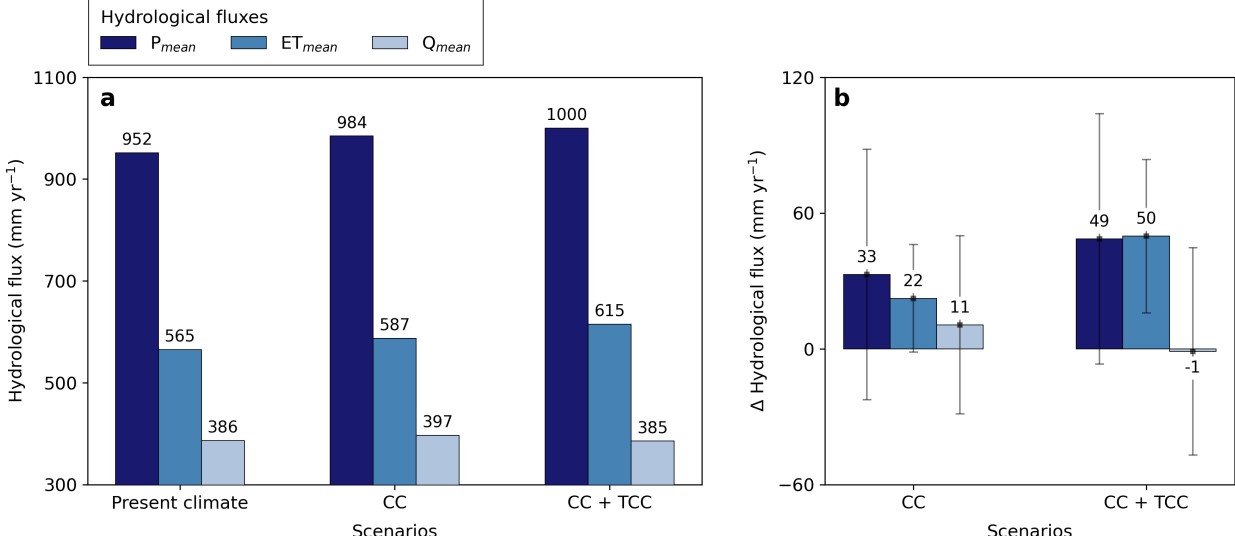

**Figure 3.** Overview of the terrestrial hydrological fluxes precipitation (P), evapotranspiration (ET), and runoff (Q) for the scenarios; present climate, climate change (CC), and climate change with tree cover change (CC + TCC). a) Average fluxes over land for each scenario, averaged over the CMIP6(-Budyko) models, in mm yr$^{-1}$. b) Average change ($\Delta$) in fluxes for each scenario relative to the present climate in mm yr$^{-1}$. The corresponding standard deviations display the variability over the CMIP6(-Budyko) models. The flux changes were calculated for each of the 30 CMIP6(-Budyko) model combinations, generating 30 $\Delta$flux values for each land grid cell from which the mean and standard deviation were computed. Note that this figure shows the weighted averages over the total land surface area and these averages are displayed as rounded values (for more details see Table A3).

For the Q flux, the spatial changes in Q due to climate change (Fig. 5a) are roughly consistent with the multi-model trends for SSP5-8.5 by Wang et al. (2022); Li and Li (2022), and the changes in Q typically follow the trends for P. However, in certain regions, such as in parts of Europe and Northern South America, the climate-driven decrease in Q seems to be more pronounced than the limited change in P, as was also shown by Cook et al. (2020) for SSP3-7.0. The difference between the P and Q trends that we found in these areas could be attributed to the elevated ET. Analyzing the impacts of an altered tree cover on Q (Fig. 5e) shows an apparent inverse relationship between Q and tree cover change, whereby an increased (decreased) tree cover generally results in a decreased (increased) Q, because of an enhanced (reduced) ET. However, the magnitude of the overall tree cover-driven impacts on Q may vary locally due to the indirect effects (altered moisture recycling) following tree cover change. Note that for both scenarios CC and CC+TCC, the variability between the CMIP6-Budyko models regarding the change in Q is most pronounced in areas that experience larger flux changes. As the high model variability is mainly concentrated in the wet lower latitudes, the variability for these areas could be (partly) explained by the large spread in P for very wet climate regions as seen in Fig. 2a. Li and Li (2022) also show a higher uncertainty in Q for the regions around the equator for SSP1-2.6, SSP2-4.5, and SSP5-8.5.



**Figure 4.** Average precipitation change ($\Delta$P; left) and evapotranspiration change ($\Delta$ET; right) in a future climate due to the impact of climate change (CC; a and b) and the combined impacts of climate change and tree cover change (CC + TCC; c and d). The hydrological changes in these figures are relative to the present climate. Figures e and f only display the impact of tree cover change (TCC; difference between CC and CC+TCC).

To analyse the spatial significance of the Q change due to a changing tree cover, we use the Wilcoxon Signed-Rank Test (Wilcoxon, 1945) (i.e. a non-parametric version of the paired samples Student's t-test) for every land grid cell with a p-value threshold of 0.05. For most of the land surface there is a significant change in Q due to tree cover change (Fig. 5f), and for some regions, such as the Sahara desert, the very small absolute change in Q is also marked as significant. We find that, for 16 % of the land surface, tree cover change (under climate change) significantly increases Q with more than $5\,\mathrm{mm\,yr^{-1}}$.







**Figure 5.** Average runoff change ($\Delta Q$: a, c, e, and f) and the corresponding standard deviation (b and d) in a future climate due to the impact of climate change (CC; a and b) and the combined impacts of climate change and tree cover change (CC + TCC; c and d). The hydrological changes in these figures are relative to the present climate. Figures e and f both display the impact of tree cover change (TCC; difference between CC and CC+TCC), whereby figure f illustrates the spatial significance for $\Delta Q$ due to TCC. The spatial significance is obtained by applying a Wilcoxon Signed-Rank Test to every land grid cell and the stippling in the figure indicates which grid cells have a significant p-value (<0.05).

As the impacts of tree cover change can amplify, mitigate, or even reverse the effects of climate change, Fig. 6 shows the spatial distribution of the separate impacts of tree cover change and climate change on Q. The figure includes nine different color combinations, illustrating whether the sign of change due to a changing tree cover aligns with or contradicts the climate change effect on Q. In addition, we use a threshold of $5\,\mathrm{mm\,yr^{-1}}$ in the figure to indicate regions with a low absolute change in Q. For approximately $6\,\%$ of the land surface, Q decreases with more than $5\,\mathrm{mm\,yr^{-1}}$ due to climate change and increases




with more than $5\,\mathrm{mm}\,\mathrm{yr}^{-1}$ following tree cover change (pink color in Fig. 6). Hence, for only a relatively small fraction of
270 the land surface, the tree cover change can mitigate the drying trend under climate change. In contrast, it is more common that
the altered tree cover can mitigate the climate-driven increase in water availability, which occurs for approximately $21\,\%$ of
the land surface (light blue color in Fig. 6). Although the altered tree cover can counteract climate change impacts in certain
regions, we also find that part of the land surface ($14\,\%$) experiences a decrease in water availability as a result of both climate
change and tree cover change effects, such as in Southern Europe and Eastern South America. For the higher latitudes on
275 the Northern Hemisphere (around $60°$ latitude) there is a clear climate-driven increase in Q whereas the effects of tree cover
change vary. We find that most areas that experience an increasing (decreasing) Q with both climate change and tree cover
change are situated in areas where there is a potential decrease (increase) in tree cover.

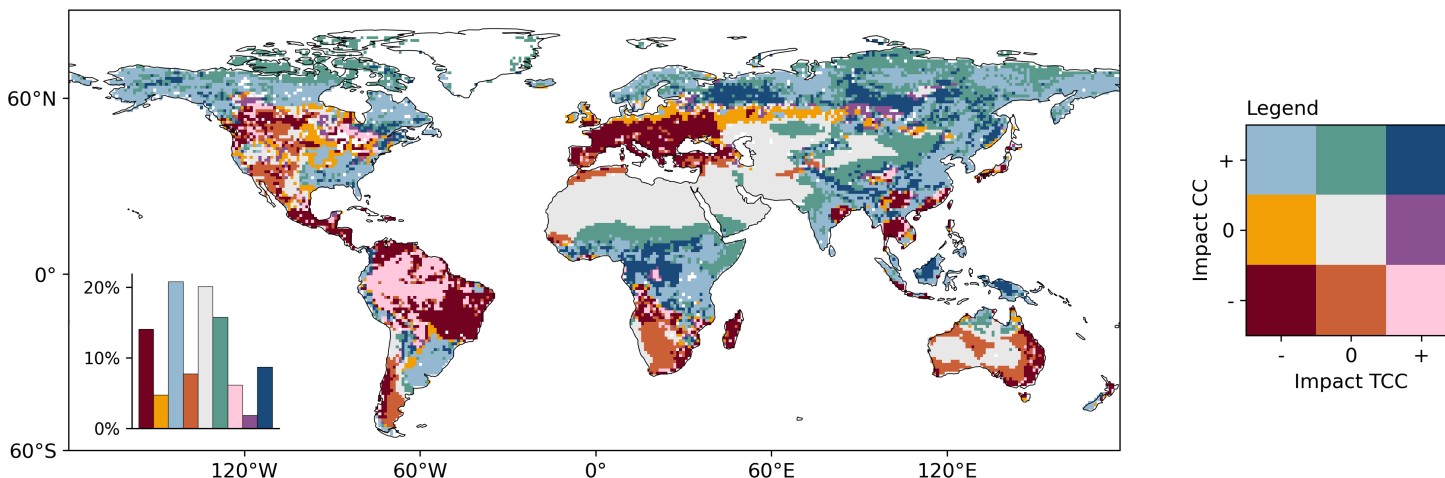

**Figure 6.** Global overview for the changes in runoff (Q) due to climate change (CC) and tree cover change (TCC). The sign of change
for Q due to CC and TCC effects can be contradictory or complementary depending on the region. The symbols '+' and '-' in the legend
correspond to a positive and negative change in Q, respectively, where these changes exceed the specified absolute threshold of 5 mm yr$^{-1}$.
The '0' in the legend represents small changes in runoff in the absolute range of 0 to 5 mm yr$^{-1}$. The legend should be read as the following;
when there is '+' for TCC and '-' for CC this means that the TCC increases the Q flux whereas CC reduces the Q flux, thus the TCC can
mitigate the negative CC effect on Q. The bar graph inside the figure shows the percentages of land surface taken up by each of the CC and
TCC combinations.

So far, we show that analysing hydrological changes on grid cell level can provide insights about the impact patterns of
280 climate change and tree cover change on the global water fluxes and water availability. However, as impact patterns can differ
extensively on a local level, the overall impact on the hydrological fluxes in a catchment remains unclear. By aggregating the
results to catchment level (Sect. 3.3), we can also obtain insights about the changes in water availability for e.g., communities,



shipping, and agriculture.

## 3.3 Catchment responses to climate change and future tree cover change

We zoom in on five catchments to take a closer look at the impacts of climate change and tree cover change on integrated hydrological fluxes. The following five catchments are selected; Yukon River basin, Mississippi River basin, Amazon River basin, Danube River basin, and the Murray-Darling River basin. These river basins are chosen since they are situated in different climate zones (Table 3), and because the catchments have a large surface area (Table 3) which is suitable for the coarse spatial resolution of our results. Furthermore, all basins encounter different impacts on Q due to changes in tree cover and climate (Fig. 7).

**Table 3.** Characteristics of the five chosen catchments in this study. The climate classification according to Köppen (1936).

| River basin | Location | Surface area$^*$ (km$^2$) | Climate zone |
|---|---|---|---|
| Yukon | North America; Canada, United States (Alaska) | 832819.3 | polar climate |
| Mississippi | North America; Canada, United States | 3240616.8 | continental climate |
| Amazon | South America | 5912922.8 | tropical climate |
| Danube | Europe | 795318.4 | temperate climate |
| Murray-Darling | Australia | 1055416.2 | arid / temperate climate |

* According to the catchment shapefiles retrieved from https://www.hydrosheds.org/products/hydrobasins

By only looking at the climate change effects, the Yukon River basin experiences an increase in P of approximately $93 \, \mathrm{mm \, yr^{-1}}$ (14.5 %) relative to the present climate, which is the largest climate-driven change in P amongst the five river basins (Fig. 7, Table A4). The pronounced increase in yearly P under climate change for this catchment agrees with the projections of Hay and McCabe (2010); Bush and Lemmen (2019), and can lead to a relatively larger increase for Q compared to ET (Hay and McCabe, 2010), which is in line with our results. This larger impact on annual Q can be attributed to a smaller increase in ET under climate change as temperatures remain relatively cold for this region (Hay and McCabe, 2010). We find that altering the tree cover enhances ET (directly) and P (indirectly) within the river basin, thereby amplifying the climate change effect for these two fluxes. The higher tree cover in the basin causes a slightly larger fraction of P to be evaporated which subsequently reduces Q and thus partly counteracts the climate-driven increase in Q. For the Wolf Creek basin, located in the Yukon catchment region, Rasouli et al. (2019) also showed that future vegetation increase can mitigate the climate-driven increase in Q. The vegetation effects could even largely offset the climate change impacts in the Wolf Creek basin as the climate and vegetation effects were found to be of similar magnitude (Rasouli et al., 2019). In contrast, we find that, for the larger catchment, the climate change impacts on water availability within the Yukon River basin are clearly dominant over the impacts of a changing tree cover.



Similarly to the Yukon River basin, the hydrological fluxes in the Mississippi River basin increase under climate change, although the extent of increase is much smaller for the latter. However, contrasting with the Yukon basin, the Mississippi catchment shows a more substantial climate-driven increase in ET compared to Q, which could be attributed to the different

dominant driving factors for ET in both regions. As the ET flux in the Mississippi River basin is primarily water-limited (Li et al., 2022a), the majority of the additional P resulting from climate change is evaporated and therefore Q encounters only a small increase of $3\,\mathrm{mm\,yr^{-1}}$ (Fig. 7, Table A4). For this catchment the dominating terrestrial moisture sources for P are situated within or to the southwest of the catchment (Benedict et al., 2020), both of which experience an increase in tree cover under scenario CC+TCC and subsequently an increase in ET. Hence, by accounting for the tree cover change, the P flux received by

the basin is further enhanced. Since more moisture is 'lost' from the catchment through increased ET than is 'gained' through enhanced P following tree cover change, the overall Q shifts to a decrease of $16\,\mathrm{mm\,yr^{-1}}$ ($-5\,\%$) relative to the present climate. Therefore, when focusing solely on the water availability, the tree cover-driven impacts are generally dominant over the impacts of climate change in the Mississippi River basin.

For the Amazon River basin, the climate change effects deviate from those in the previous discussed catchments as there is a decrease in P under pathway SSP3-7.0, in line with Almazroui et al. (2021); Cook et al. (2020). In most of the Amazon region, ET is limited by the available energy (Li et al., 2022a) and therefore the strong increase in temperature for this region under SSP3-7.0 (Almazroui et al., 2021) can enhance ET. Under scenario CC+TCC, the Amazon River basin experiences a widespread (slight) tree cover decrease due to reduced water availability and extended periods of drought, leading to higher

tree mortality (Tavares et al., 2023; Wey et al., 2022). However, the Amazon also contains areas with an increasing tree cover, for example along the southern catchment boundary, that offset the decreases on catchment scale, resulting in an overall tree cover increase of $4\,\%$ (Fig. 7). This compensation effect on catchment scale also occurs for ET since there is an overall tree cover-driven increase in ET regardless of the widespread reduction following tree cover change (Fig. 4f). According to Tuinenburg et al. (2020), the Amazon River basin has the highest local moisture recycling of our five selected catchments ($63\,\%$

of the locally evaporated moisture rains out within the basin). Hence, the increasing tree cover within (and upwind of) the Amazon basin can enhance ET (directly) and P (indirectly) on catchment level, thereby amplifying and mitigating the climate change impacts on these fluxes, respectively. The changing tree cover can also enhance Q with $2\,\mathrm{mm\,yr^{-1}}$ ($0.3\,\%$) in the basin. According to Guimberteau et al. (2017), deforestation effects in a future climate could substantially mitigate the climate-driven decreases for Q at a sub-basin scale in the Amazon. However, we find that, on catchment level, the tree cover change only has

a limited effect on Q. Therefore, the climate change impacts dominate over the effects of tree cover change regarding the water availability in the Amazon River basin.

In the Danube River basin the hydrological trends under climate change mirror those shown for the Amazon River basin as both catchments experience a decreasing P and Q while ET increases (Fig. 7). Interestingly, this decreasing P and Q that we

find for the Danube basin is in contrast with the enhanced P and subsequently increased Q under climate pathways RCP2.6 and RCP8.5 shown by Probst and Mauser (2023). However, it should be noted that in the study of Probst and Mauser (2023) as



**Figure 7.** Mean change for the hydrological fluxes precipitation (P), evapotranspiration (ET), and runoff (Q) due to climate change (CC) and climate change with tree cover change (CC + TCC), relative to the present climate. The changes are in mm yr$^{-1}$. On the left the Yukon (North America; Canada, United States (Alaska)), Mississippi (North America; Canada, United States), and Amazon (South America) River basins are displayed with the corresponding flux changes and on the right the Danube (Europe) and Murray-Darling (Australia) River basins are shown with the corresponding flux changes. The river basins are indicated with a dark blue line. The TCC$_{mean}$ value in the upper left of each bar plot shows the average tree cover change in the catchments and the Q$_{CC+TCC}$ shows the total Q in the catchment for research scenario CC + TCC. Note that the values presented in the bar graphs are rounded, for more details see Table A4.

well as in our study (Table A4), there can be large uncertainties for the projections of future P. Accounting for the changing tree cover, the Danube River basin experiences a tree cover increase of 37.6 % and thus has the largest tree restoration potential over the five river basins. In addition, the dominant terrestrial moisture sources of the Danube basin (Central and Eastern Europe

as well as the catchment itself (Ciric et al., 2016)) also experience an enhanced tree cover. As a result, the P flux received by the catchment increases (indirectly) and consequently shifts the climate-driven P decrease to an overall increase. Following the enhanced tree cover, a larger fraction (71.5 %) of the incoming P is evaporated and thus a smaller P fraction ends up in the Q flux, which contributes to an amplification of the negative climate change effect on Q. Overall, we find that the impacts of





climate change and tree cover change are of similar magnitude for the water availability in the Danube River basin.


Last, focusing on the Murray-Darling River basin, we see that climate change causes a decrease for all hydrological fluxes, contrasting to the other catchments discussed. This deviation could be attributed to the more arid climate conditions within the basin (Table 3). The decrease in P resulting from climate change can subsequently lead to reduced ET (Fig. 7) as the ET in this catchment is constrained by water availability (Li et al., 2022a). Including the (limited) tree cover increase (scenario

CC+TCC) in the Murray-Darling River basin can shift the ET from a climate-driven decrease to a small increase. However, the incoming P is enhanced only slightly since the region shows a relatively low moisture recycling whereby only $11\%$ of the locally evaporated moisture again precipitates in the basin (Tuinenburg et al., 2020). In addition, the catchment receives little precipitation recycled from upwind terrestrial regions (Holgate et al., 2020), indicating that upwind changes in tree cover have minimal impact on the P received by this catchment. The small increase in P due to a locally enhanced tree cover cannot

compensate for the additional moisture 'loss' from the basin through enhanced ET, thereby amplifying the impact of climate change on Q. Overall, we find that the climate change impacts are dominant over the effects of tree cover change for the water availability in the Murray-Darling River basin.

In conclusion, we show for five large catchments distributed over different climate zones that the impacts of climate change

and potential future tree restoration can deviate substantially on a regional scale. These results provide insights into how climate-driven changes in tree cover or tree restoration measures can enhance or offset unwanted climate change effects in each catchment. However, it should be kept in mind that these results correspond with noteworthy uncertainties due to divergent CMIP6 model projections and the use of various Budyko models.

## 4 Discussion of methodology

This study provides a first estimate of the impacts of tree cover change under climate change on global hydrological fluxes over land. To study the combined and separate impacts of tree cover change under climate change, we took an interdisciplinary approach combining state-of-the-art datasets and methods following the methodology from Hoek van Dijke et al. (2022), adapted for future climate scenarios. We would like to stress that other studies also applied the Budyko method and UTrack dataset under different climate and land cover scenarios Kazemi et al. (2019); Tuinenburg et al. (2022); Teuling et al. (2019);

Li et al. (2022a). In the discussion below we reflect on the implications of the methodological constraints and their potential impacts on the results.

### 4.1 Implications of the SSP3-7.0 pathway, tree cover change map, and Budyko method

The SSP3-7.0 climate pathway was selected, because it enabled us to study strong climate change effects, without stretching our methods too much towards unknown conditions. Both the tree cover map and Budyko models rely on the change for time

principle: in a warmer climate we will find a similar distribution of trees as in the current climate, however in a different spatial



region. For a stronger climate change scenario, more frequent extrapolation outside of current conditions would introduce uncertainties in the future potential tree cover map and make the use of the Budyko framework more uncertain. However, some uncertainties and limitations remain.

The SSP3-7.0 pathway describes a resource intensive world with, unlike the other SSP scenarios, a strong reduction in tree cover in the coming decades (Hurtt et al., 2020; Shiogama et al., 2023). The land use changes corresponding to SSP3-7.0 can affect temperature and precipitation extremes, whereby the land-cover impacts may be more pronounced on a regional level compared to the global level (Hong et al., 2022). In our study, we do not consider the climatic impacts of these land-cover changes (which can vary between CMIP6 models). Hence, this assumption may result in the under- or overestimation of climate change effects on a regional level.

To create the future potential tree cover map for SSP3-7.0 (Roebroek, 2023), no feedback between changing tree cover and the climate in SSP3-7.0 was included. By using the potential tree cover map for SSP3-7.0 and assuming large-scale tree cover change, we deviate from this climate pathway, which subsequently should alter climate characteristics and therefore the future potential tree cover. Additionally, the potential tree cover map describes the tree cover that could be established given certain climate conditions. However, three decades would not suffice to reach this level of tree cover everywhere, especially in areas

that currently do not contain trees. Furthermore, we assume a static tree cover, and do not consider temporal variability in water fluxes that result from e.g., forest disturbance and forest succession stages (Goeking and Tarboton, 2020; Teuling and Hoek van Dijke, 2020). To explicitly model all feedbacks between climate, tree cover, and tree growth one would need a fully integrated earth system model.

The Budyko models used were calibrated under current climate conditions, but the evapotranspiration to precipitation ratio

(ET/P) could change in the future. The parameterisation of the models reflects the catchment-integrated effects of differences in interception, plant available water, evaporation, water use efficiency and soil water storage capacity. Some of those characteristics could potentially change the evapotranspiration to precipitation ratio (ET/P) under future climate conditions. For example, the Budyko models do not take into account the $CO_2$ fertilization effect on reduced surface conductance and increased vegetation greenness (Zhu et al., 2016), which changes the albedo and water use efficiency (Bala et al., 2006). Also, the Budyko

vegetation parameters are sensitive for, among others, tree species and short vegetation coverage (Chen et al., 2021; Liu et al., 2018; Ding et al., 2022), and the hydrological impacts of forest restoration are therefore highly dependent on the characteristics of forest restoration (e.g. (monoculture) plantations versus natural regrowth of vegetation, and coniferous tree species versus deciduous species).

## 4.2 Implications of missing the feedback of climate change and tree cover change on recycling

The UTrack dataset used in this study is based on ERA5 reanalysis data and thus represents moisture recycling for current climate and land cover characteristics. However, in this study we apply the UTrack dataset for a future climate and therefore we do not account for the impact of climate change and tree cover change on moisture recycling. So far, Findell et al. (2019) and Staal et al. (2024a) determined moisture recycling ratios globally in a future climate, both for one climate model, and found that continental moisture recycling ratios decrease with 2–3 % for each degree of warming. Or differently said, the land-to-



land water vapor transport decreases. This signal is explained by the fact that evapotranspiration over land is moisture-limited, allowing oceans to have a relatively larger role in the hydrological cycle. Although included in the analyses, Findell et al. (2019) and Staal et al. (2024a) did not discuss in detail the impacts of changes in future circulation on moisture recycling. There are clear indications that climate warming impacts atmospheric circulation, possibly resulting in a poleward shift of the Hadley cells and storm tracks (Shaw, 2019; Francis and Skific, 2015; Vecchi et al., 2006). Future studies applying moisture tracking on future climate simulations will provide regional insights on the impact of shifting weather patterns on moisture recycling.

Projecting the reported decrease in future recycling on this study would mean that less precipitation due to additional evapotranspiration (due to tree cover change) will return over land, which will thus decrease the water availability. This indicates that under climate change and tree cover change (scenario CC+TCC), the runoff may become lower compared to the results in our study. Findell et al. (2019) also shows that under climate change, the contribution of oceanic evaporation to precipitation over land is larger, as was confirmed for the Mississippi River basin by Benedict et al. (2020). It should be noted that the higher evaporation rates over the ocean and the resulting feedback of increased precipitation over land are part of the CMIP6 datasets for a future climate, and these climate change effects are thus included in this study.

Moreover, this research does not account for tree cover change feedbacks on moisture recycling via changes in e.g. the surface albedo, cloud cover, atmospheric carbon dioxide concentrations, surface temperatures, length of moisture transport pathways, and global circulation. These feedbacks are complex and can be contrasting, depending on the location of the land-use change, and the same holds for representing those feedbacks in earth-system models (Portmann et al., 2022; De Hertog et al., 2023; King et al., 2024). Trends of general wetting and increased local moisture recycling are shown by De Hertog et al. (2023) following afforestation in two earth-system models, indicating that regions with increasing evapotranspiration due to an enhanced tree cover will receive more precipitation than currently estimated, thus limiting the decrease in local water availability. The afforestation impacts found by De Hertog et al. (2023) are opposite to the effect of climate change decreasing the local recycling, however those opposite effects are not quantified.

To conclude, the approach of this study allows to disentangle the impacts of climate change and future tree cover change on hydrological fluxes, but does not include all feedbacks in the earth system in a future climate. To include all those feedbacks, one would need to run coupled earth system models including tree cover change such as done for current climate by Portmann et al. (2022); De Hertog et al. (2023). Similar model simulations would be needed for future climate conditions and future potential tree cover, ideally for multiple earth system models that actively couple the biosphere and atmosphere. Such simulations are not widely available, to our knowledge a global simulation by King et al. (2024) is available for one earth system model, and a regional simulation by Buechel et al. (2024) for the UK using one convection permitting model. Besides, previous research shows that coupled models do not agree on the implementation of land-atmosphere processes in earth system models, resulting in uncertainty in the sensitivity of vegetation to changing water availability (Denissen et al., 2022; Li et al., 2022b; Baker et al., 2021). Overall, earth-system model studies and data-driven studies, like this one, face different sources of uncertainty and therefore have specific strengths and shortcomings. We believe that our study complements earth-system model studies by



contributing to the diversity of methodological approaches presented in scientific literature. Different approaches can improve our understanding of uncertainties and thus enable the most robust scientific progress, which is important when advising society and policymakers. Given the constraints mentioned above, our study provides a first estimate of water availability under climate change and future tree cover change.

## 5 Conclusions

In this study we analysed the impacts of climate change and global tree cover change on the terrestrial precipitation, evapotranspiration, and runoff. To do so, we took an interdisciplinary approach and combined multiple datasets and models. The hydrological fluxes under climate change and future tree cover change were compared to present climate fluxes to analyse the magnitude and direction of changes in water availability.

We find that, globally averaged, climate change and large-scale tree cover change can exert similar absolute impacts on
runoff. However, these impacts have opposite signs and thus generate a net limited effect relative to the present climate. Following climate change, there is an overall larger increase in precipitation than evapotranspiration over land, resulting in enhanced runoff. On the contrary, large scale tree restoration (15.5%, see Fig. 1) will increase evapotranspiration more than the recycled precipitation (which partly rains out over water bodies) and thus decreases the runoff.

While the average impact on global runoff is limited, the effects of tree cover change and climate change on runoff can be
substantial on a regional scale, resulting in enhanced and decreased trends in local runoff up to $100\,\mathrm{mm\,yr^{-1}}$. For example, the Amazon region will experience a strong climate-driven decrease in runoff, whereas the related reduction in tree cover could potentially mitigate these impacts. The drying trend under climate change also applies to Southern Europe, however, in this region the natural or human-induced forestation would exacerbate rather than mitigate the drying. We find that $14\,\%$ of the global land area could experience a pronounced decrease in water availability due to both climate change and tree cover
change. In contrast, the higher latitudes on the Northern Hemisphere (around $60°$ latitude) show climate-driven enhances in runoff and varying effects of tree cover change. Overall, for approximately $16\,\%$ of the land surface, tree cover changes could increase water availability (in)directly with more than $5\,\mathrm{mm\,yr^{-1}}$. Analysing the river basin results in this study shows that the climate change and tree cover change effects can diverge substantially per catchment, whereby four out of five catchments encounter dominant impacts of climate change on the regional water availability.

This is the first study to disentangle the effects of climate change and large-scale tree cover change regarding the future water availability on a global scale and for selected river catchments. We show that climate-driven or human-induced changes in tree cover can mitigate as well as exacerbate climate-induced drying or wetting trends. Ecosystem restoration projects should consider these long-term hydrological effects to limit unintended reductions for local, downstream, and downwind water availability. As a next step, we recommend the use of local coupled modelling studies whereby e.g., different afforestation
scenarios can be implemented in a regional weather model under future climate conditions, such as (Buechel et al., 2024) for the UK. Such studies would enable the analysis of direct local feedbacks and sensitivities of tree cover changes to evaporation



and precipitation, based on local atmospheric conditions. Hence, these local studies could address and verify the hydrological responses that we find in our study and thereby guide local and global forest restoration projects.

*Data availability.* The CMIP6 climate simulations are publicly available from https://esgf-node.llnl.gov/search/cmip6/. Potential evapotran-
spiration (PET) data derived from the CMIP6 simulations is provided by Bjarke et al. (2023). The UTrack dataset is publicly available from https://doi.pangaea.de/10.1594/PANGAEA.912710. The tree cover dataset for the present climate is provided by Hansen et al. (2013) and the future potential tree cover datasets are provided by Roebroek (2023).

*Author contributions.* FE carried out the study and created all figures. The idea of this study was conceived by IB and AHvD. FE, IB, and AHvD wrote the first draft of the paper. All authors interpreted the results, contributed to the discussion and were involved in writing the
final paper.

*Competing interests.* The authors declare no competing interests.

*Acknowledgements.* AHvD acknowledges funding by the German Research Foundation (Emmy Noether grant 391059971). We would like to thank two anonymous reviewers for their feedback on an earlier version of this manuscript.



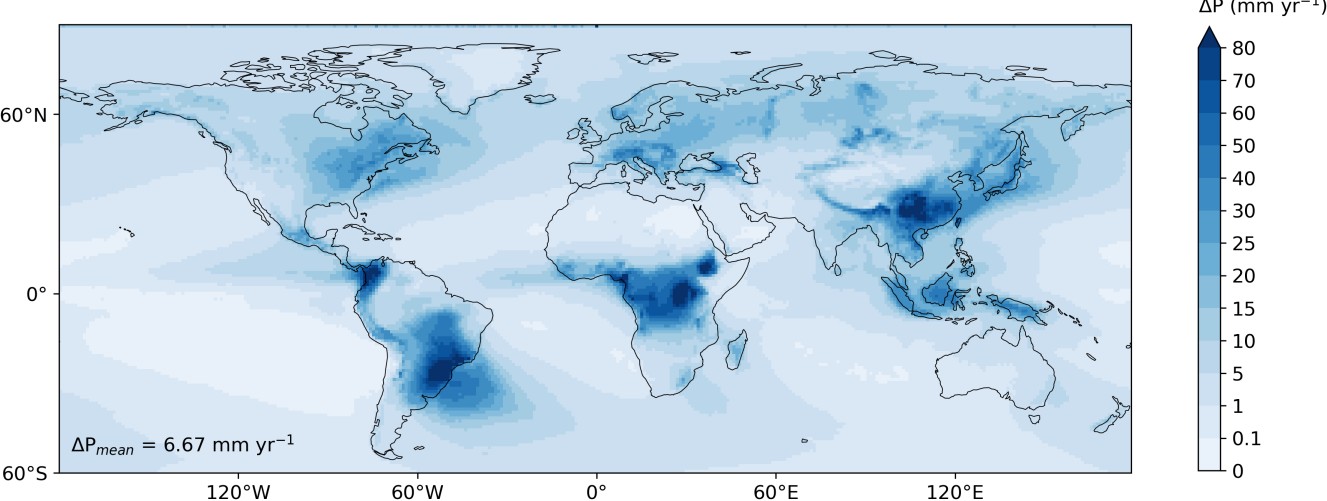

**Figure A1.** Average global change in precipitation (ΔP) in a future climate (2041-2060) due to the indirect impacts of a changing tree cover which affects the moisture recycling. This figure shows the ΔP averaged over the CMIP6-Budyko model combinations and the moisture recycling change was applied twice.



**Table A1.** Datasets used in this study with corresponding characteristics and the sources from which these datasets can be obtained. The datasets with an asterisk symbol (*) were retrieved for five models selected from phase 6 of the Coupled Model Intercomparison Project (CMIP6). The datasets were used in the following scenarios; present climate, climate change, climate change with tree cover change (TCC), climate change with TCC and moisture recycling change (MRC) applied once, and climate change with TCC and MRC applied twice. Note that scenario 'Climate change' is presented as the 'CC' scenario outside of the appendix, while 'Climate change + TCC + 2x MRC' is presented as the 'CC + TCC' scenario.

| Data type | Dataset (climate pathway) | Time period (temporal resolution) | Source | Used for research scenario |
|---|---|---|---|---|
| CMIP6 climate data | Precipitation* (Historical pathway) | 1985–2014 (monthly) | https://esgf-node.llnl.gov/ search/cmip6/ | Present climate |
| | Potential evapotranspiration* (Historical pathway) | 1985–2014 (monthly) | (Bjarke et al., 2023) | |
| | Precipitation* (SSP3-7.0) | 2035–2064 (monthly) | https://esgf-node.llnl.gov/ search/cmip6/ | Climate change, Climate change + TCC, Climate change + TCC + 1x MRC, Climate change + TCC + 2x MRC |
| | Potential evapotranspiration* (SSP3-7.0) | 2035–2064 (monthly) | (Bjarke et al., 2023) | |
| Tree cover data | Tree cover present climate | 2000 | (Hansen et al., 2013) | Present climate, Climate change |
| | Potential tree cover future climate* (SSP3-7.0) | 2041–2060 | (Roebroek, 2023) | Climate change + TCC, Climate change + TCC + 1x MRC, Climate change + TCC + 2x MRC |
| Moisture tracking data | UTrack moisture trajectories at 1°x1° resolution | 2008–2017 (monthly) | https://doi.pangaea.de/10. 1594/PANGAEA.912710 | Climate change + TCC + 1x MRC, Climate change + TCC + 2x MRC |

* The datasets are retrieved for five selected CMIP6 models; CMCC-ESM2, INM-CM5-0, IPSL-CM6A-LR, MIROC6, and UKESM1-0-LL.





**Table A2.** The six Budyko models used in this study for which the evapotranspiration (ET) and runoff (Q) fluxes are calculated from the CMIP6 precipitation (P) and potential evapotranspiration (PET) datasets. The $\omega$ for these models is calibrated for forest and non-forest vegetation, based on yearly mean streamflow or lysimeter data from various climatic regions. Streamflow Q is calculated as $\frac{Q}{P} = 1 - \frac{ET}{P}$.

| Model | Functional form | Calibrated $\omega$ | Details |
|---|---|---|---|
| 1 | $$\frac{ET}{P} = \frac{1 + \omega\frac{PET}{P}}{1 + \omega\frac{PET}{P} + \frac{P}{PET}}$$ | $\omega_g = 0.5$ <br> $\omega_f = 2.0$ | Model developed and calibrated by Zhang et al. (2001). Calibrated using 240 global river basins. $\omega_f$ is calculated using natural and plantations data, and $\omega_g$ is calibrated using grassland and cropland data. |
| 2 | $$\frac{ET}{P} = \frac{1 + \omega\frac{PET_z}{P}}{1 + \omega\frac{PET_z}{P} + \frac{P}{PET_z}}$$ | $\omega_g = 0.5$ <br> $\omega_f = 2.0$ | As Model 1. $PET$ is also calibrated ($PET_z$). $PET_{z,grass} = 1100$, $PET_{z,trees} = 1410$ |
| 3 | $$\frac{ET}{P} = 1 + \frac{PET}{P} - (1 + (\frac{PET}{P})^\omega)^{\frac{1}{\omega}}$$ | $\omega_g = 2.55$ <br> $\omega_f = 2.84$ | Model developed by Fu in 1981, calibrated by Zhang et al. (2004). Calibrated based on 200 Australian and 270 worldwide river basins. $\omega_f$ and $\omega_g$ are calibrated using river basins $\leq 75\%$ forest and grassland cover. |
| 4 | $$\frac{ET}{P} = 1 + \frac{PET}{P} - (1 + (\frac{PET}{P})^\omega)^{\frac{1}{\omega}}$$ | $\omega_g = 2.28$ <br> $\omega_f = 2.83$ | Model developed by Fu in 1981, calibrated by Zhang et al. (2004). Calibrated with 1420 river basins with forest ($\omega_f$), and grassland and cropland ($\omega_g$) cover. |
| 5 | $$\frac{ET}{P} = 1 + \frac{aPET}{P} - (1 + (\frac{aPET}{P})^\omega)^{\frac{1}{\omega}}$$ | $\omega_g = 1.7$ <br> $\omega_f = 3.1$ | Model developed by Fu in 1981, calibrated by Teuling et al. (2019). Calibrated based on European Lysimeter data. Teuling et al. (2019) introduced the adjusted potential evaporation ($aPET = 1.6PET$) to account for lysimeter observations above the energy line. |
| 6 | a) $$\frac{ET}{P} = 1 - exp(-\frac{\omega PET}{P})$$ <br><br> b) $$\frac{ET}{P} = \omega\frac{PET}{P}tanh((\omega\frac{PET}{P})^{-1})$$ <br><br> c) $$\frac{ET}{P} = \frac{1}{(1 + (\omega\frac{PET}{P})^{-2})^{0.5}}$$ <br><br> d) $$\frac{ET}{P} = (\frac{PET}{P}(1 - exp(-\omega\frac{PET}{P}))tanh(\frac{P}{PET}))^{0.5}$$ <br><br> e) $$\frac{ET}{P} = \frac{1 + \omega\frac{PET}{P}}{1 + \omega\frac{PET}{P} + \omega + \frac{P}{PET}}$$ | $\omega_g = 0.977$ <br> $\omega_f = 1.248$ <br><br> $\omega_g = 0.767$ <br> $\omega_f = 0.910$ <br><br> $\omega_g = 0.831$ <br> $\omega_f = 1.025$ <br><br> $\omega_g = 0.762$ <br> $\omega_f = 1.125$ <br><br> $\omega_g = 0.682$ <br> $\omega_f = 1.404$ | Mean $ET$ calculated from five Budyko equations of: a) Schreiber, b) Ol'DeKop, c) Turc, d) Budyko, and e) Zhang. Oudin et al. (2008) introduced $\omega$ in these equations to capture the vegetation effects. The formulas are calibrated using data from 1508 river basins in United States, United Kingdom, Sweden and France. |





**Table A3.** Overview of the global terrestrial hydrological flux values for precipitation (P), evapotranspiration (ET), and runoff (Q) for the scenarios; present climate, climate change, climate change with tree cover change (TCC), climate change with TCC and moisture recycling change (MRC) applied once, and climate change with TCC and MRC applied twice. The mean of the variables represents the mean over the CMIP6(-Budyko) models and the corresponding standard deviations display the variability over the CMIP6(-Budyko) models. Note that scenario 'Climate change' is presented as the 'CC' scenario outside of the appendix, while 'Climate change + TCC + 2x MRC' is presented as the 'CC + TCC' scenario.

| Variable | Units | Present climate land total | Change in variable relative to present climate | | | |
|---|---|---|---|---|---|---|
| | | | Climate change | Climate change + TCC | Climate change + TCC + 1x MRC | Climate change + TCC + 2x MRC |
| Tree cover mean | % | 23.1 | +0.0 | +15.5 | +15.5 | +15.5 |
| P mean $\pm$ std | mm yr$^{-1}$ | $951.5 \pm 198.7$ | $+32.8 \pm 55.3$ | $+32.8 \pm 55.3$ | $+45.8 \pm 55.3$ | $+48.6 \pm 55.3$ |
| ET mean $\pm$ std | | $565.2 \pm 103.1$ | $+22.3 \pm 23.8$ | $+44.6 \pm 33.5$ | $+48.8 \pm 33.8$ | $+49.7 \pm 33.9$ |
| Q mean $\pm$ std | | $386.3 \pm 158.8$ | $+10.5 \pm 39.3$ | $-11.8 \pm 45.0$ | $-3.0 \pm 45.7$ | $-1.1 \pm 45.8$ |



**Table A4.** Overview of the catchment hydrological flux values for precipitation (P), evapotranspiration (ET), and runoff (Q) for the scenarios; present climate, climate change, climate change with tree cover change (TCC), climate change with TCC and moisture recycling change (MRC) applied once, and climate change with TCC and MRC applied twice. The values of the variables represent the mean over the CMIP6(-Budyko) models and the corresponding standard deviations, the latter of which display the variability over the CMIP6(-Budyko) models. Note that scenario 'Climate change' is presented as the 'CC' scenario outside of the appendix, while 'Climate change + TCC + 2x MRC' is presented as the 'CC + TCC' scenario.

| Catchment | Variable | Units | Present climate land total | Change in variable relative to present climate | | | |
|---|---|---|---|---|---|---|---|
| | | | | Climate change | Climate change + TCC | Climate change + TCC + 1x MRC | Climate change + TCC + 2x MRC |
| Amazon | Tree cover | % | 82.7 | +0.0 | +4.0 | +4.0 | +4.0 |
| | P ± std | mm yr$^{-1}$ | 2191.4 ± 457.0 | −38.3 ± 105.6 | −38.3 ± 105.6 | −25.0 ± 105.6 | −21.5 ± 105.6 |
| | ET ± std | | 1325.4 ± 218.4 | +15.1 ± 42.2 | +23.8 ± 49.6 | +28.1 ± 49.4 | +29.3 ± 49.3 |
| | Q ± std | | 866.0 ± 376.0 | −53.4 ± 80.8 | −62.1 ± 86.3 | −53.1 ± 86.4 | −50.8 ± 86.4 |
| Danube | Tree cover | % | 30.8 | +0.0 | +37.6 | +37.6 | +37.6 |
| | P ± std | mm yr$^{-1}$ | 778.0 ± 140.9 | −7.4 ± 39.1 | −7.4 ± 39.1 | +15.2 ± 39.1 | +20.4 ± 39.1 |
| | ET ± std | | 503.8 ± 68.3 | +18.8 ± 21.0 | +54.6 ± 30.0 | +64.6 ± 30.3 | +66.9 ± 30.4 |
| | Q ± std | | 274.1 ± 113.5 | −26.2 ± 29.9 | −62.0 ± 36.1 | −49.3 ± 36.5 | −46.5 ± 36.7 |
| Mississippi | Tree cover | % | 21.6 | +0.0 | +21.5 | +21.5 | +21.5 |
| | P ± std | mm yr$^{-1}$ | 929.9 ± 79.8 | +34.7 ± 40.4 | +34.7 ± 40.4 | +47.9 ± 40.4 | +50.4 ± 40.4 |
| | ET ± std | | 603.4 ± 54.1 | +31.8 ± 16.0 | +60.9 ± 25.5 | +65.8 ± 25.9 | +66.7 ± 26.0 |
| | Q ± std | | 326.6 ± 68.9 | +2.9 ± 30.5 | −26.2 ± 35.7 | −17.9 ± 36.3 | −16.4 ± 36.4 |
| Murray-Darling | Tree cover | % | 8.8 | +0.0 | +7.1 | +7.1 | +7.1 |
| | P ± std | mm yr$^{-1}$ | 737.7 ± 115.1 | −23.2 ± 34.7 | −23.2 ± 34.7 | −20.6 ± 34.7 | −20.3 ± 34.7 |
| | ET ± std | | 551.4 ± 67.1 | −5.9 ± 21.0 | +1.6 ± 23.4 | +2.9 ± 23.3 | +3.1 ± 23.3 |
| | Q ± std | | 186.3 ± 62.3 | −17.3 ± 16.3 | −24.8 ± 18.0 | −23.6 ± 18.0 | −23.4 ± 18.0 |
| Yukon | Tree cover | % | 36.0 | +0.0 | +19.7 | +19.7 | +19.7 |
| | P ± std | mm yr$^{-1}$ | 640.8 ± 104.6 | +92.8 ± 40.5 | +92.8 ± 40.5 | +101.4 ± 40.5 | +102.1 ± 40.5 |
| | ET ± std | | 299.5 ± 105.8 | +40.2 ± 21.6 | +54.2 ± 28.1 | +55.8 ± 29.1 | +56.0 ± 29.1 |
| | Q ± std | | 341.3 ± 141.6 | +52.6 ± 26.7 | +38.6 ± 32.1 | +45.5 ± 33.0 | +46.1 ± 33.1 |



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
