# Peer review of "Can large-scale tree cover change negate climate change impacts on future water availability?"

_EGUsphere, 2024_

## Referee Comment (RC1)

**HESS Manuscript # https://doi.org/10.5194/egusphere-2024-2015**

Title:   Can large-scale tree cover change negate climate change impacts on future water availability?

Authors: Engel et al.

**Review**

This manuscript describes an initial evaluation of how possible future tree planting (up to the potential carrying capacity) would interact with climate change to change water availability. The study is a useful first estimate of an important practical problem since tree planting is often promoted as being a suitable mitigation strategy.

The topic is suitable for the journal and the paper is very well written and easy to follow. I had very few substantive comments.

As an initial study of a complex phenomenon it has many drawbacks but to the credit of the authors many of those are described in detail in section 4.

One aspect of the project design/methodology that was not commented upon is the inclusion of the Budyko process to separate P into Q and ET. I did not understand why this step was included since the climate model projections include ET and Q and you could use those directly. (See Roderick et al 2015; Milly & Dunne 2017). (Also see point 3 below.)

I think it might be useful to extend the discussion a little further to give full context. The idea of planting trees to their "potential" and then assessing the hydrologic changes is challenging but useful. Overall the global impacts of tree planting (or climate change) on runoff were small (Fig. 3) but could be important locally. Imagine we actually did plant all of those trees. The other impacts are on biodiversity (likely increase) but the big impact would be on agricultural production which would decrease (by a lot). Something should be said about this in the discussion.

**Details**

1. Line 16. "…. Water availability with more than 5 mm yr$^{-1}$." I did not understand what the 5 mm yr-1 was referring to?

2. Line 34. Typo.  … increased **by** more

3. Line 403-404. Yes, the Budyko models do not explicitly account for that but the climate models do (see Yang et al 2019, Hydrologic implications of vegetation response to elevated $CO_2$ in climate projections, Nature Climate Change). This begs the question of why use the Budyko models to split the P between ET and Q. You could use climate model output directly and avoid the Budyko step in the methodology.

**References Cited**

Milly, P. C. D. and Dunne, K. A.: A Hydrologic Drying Bias in Water-Resource Impact Analyses of Anthropogenic Climate Change, Journal of the American Water Resources Association, 10.1111/1752-1688.12538, 2017.

Roderick, M. L., Greve, P., and Farquhar, G. D.: On the assessment of aridity with changes in atmospheric CO2, Water Resources Research, 51, 5450-5463, 10.1002/2015wr017031, 2015.

**Michael L. Roderick, 1/10/2024**

---

## Author Response (AR1)

**Editor**

Dear authors,

as you have seen, we have received very detailed and thoughtful comments by three reviewers. The reviews diverge in their assessment of the added value of your analysis. Two reviewers are very positive while the third reviewer is not convinced. I thank you for the detailed replies to all reviewers and your additional perspective on the intention of your manuscript.

Overall, I share the positive evaluation of reviewers #1 and #3 and I believe that your work has some interesting insights to offer. However, I also share some of the concerns of reviewer #2. I agree with your view that the mechanisms involved are complex, intertwined and highly problematic to disentangle and that analyses from different perspectives may be helpful to gain new insights. Yet, this contrasts with partly quite sweeping statements and generalizations you make throughout the manuscript.

Given the detailed and very honest list of limitations in Section 4, which I highly appreciate, it will strengthen your manuscript if you provided more nuanced and less assertive descriptions in Section 3 and in the conclusions (Section 5). In other words, the presentation and discussion of your results as well as the conclusions need to be toned down and communicated much more in the light of the assumptions made and the uncertainties following from that. As a simple example, it is stated that:"[...] the Danube River basin experiences a tree cover increase of 37.6 % [...]". Well, no. There is no data to support that. What you probably meant to say is:" [...] the Danube River basin has an estimated potential for a 37.6% tree cover increase [...]". These are at first glance minor issues but imprecise formulations like that contribute to severe misunderstandings.

Furthermore, Section 4, reads like a mere list of potential sources of uncertainty. It will be good to give this a bit more thought to also provide thoughts on what the effect of these uncertainties may have on your conclusions. For example, you state that: "Budyko models do not take into account the CO2 fertilization effect on reduced surface conductance and increased vegetation greenness (Zhu et al., 2016), which changes the albedo and water use efficiency [...]". Great. But so what? Give the reader some more insight on how this may propagate through your results.

I thus encourage you to incorporate all reviewer comments and invest some more effort in revising the text so as to make sure to precisely communicate your findings and associated uncertainties, which will help the reader to place your work into a meaningful context.

I am looking forward to receiving a revised version of your manuscript,
best regards,
Markus Hrachowitz

We thank Dr. Markus Hrachowitz for the quick response to our manuscript and the detailed constructive feedback comments to improve the manuscript. We have incorporated the comments by the editor as well as the reviewers in the revised version of our manuscript.
In various sections of the manuscript, we now provide more nuanced description of the results and conclusions. A few examples are:

- L224: *We estimate the mean global ET to enhance with 28 +- 19 mm yr-1 (4.7%) relative to scenario CC following an average increase of 15.5% in global tree cover.*
- L230: *In addition, the projected change in tree cover can generate a shift in Q from a climate-driven net increase in Q to a slight net decrease.*

- L282: (...) *the change in tree cover can mitigate the drying trend under climate change. In contrast, it appears to be more common for tree restoration to mitigate the climate-driven increase* (...).
- L346 *In contrast, we find only a limited net impact of tree cover change on Q in the Amazon River basin, suggesting that climate change has a more pronounced influence on water availability compared to tree cover change in this region.*

Furthermore, in the discussion, we now provide examples of how the uncertainties could propagate into our results. A few examples are:

L402: *For example, the large-scale deforestation in tropical regions presented in SSP3-7.0 could decrease the regional mean and extreme precipitation (Hong et al., 2022). Hence, the precipitation increase in tropical Africa, which we attribute solely to climate change, may actually consist of a stronger increase driven by climate change combined with a decrease driven by deforestation, meaning that for this region the climate change effects could be underestimated.*

L427: *Although these opposing effects are of similar importance on a global scale, the regional effect can be highly positive or negative (Zhang et al., 2022), and therefore, the Budyko parameters and our calculation of ET, will be non-uniformly affected.*

References:

Hong, T., Wu, J., Kang, X., Yuan, M., and Duan, L.: Impacts of Different Land Use Scenarios on Future Global and Regional Climate Extremes, Atmosphere, 13, https://doi.org/10.3390/atmos13060995, 2022

Zhang, X., Zhang, Y., T. J., Ma, N., and Wang, Y.-P.: CO2 fertilization is spatially distinct from stomatal conductance reduction in controlling ecosystem water-use efficiency increase, Environmental Research Letters, https://doi.org/10.1088/1748-9326/ac6c9c

**Reviewer 1**

This manuscript describes an initial evaluation of how possible future tree planting (up to the potential carrying capacity) would interact with climate change to change water availability. The study is a useful first estimate of an important practical problem since tree planting is often promoted as being a suitable mitigation strategy. The topic is suitable for the journal and the paper is very well written and easy to follow. I had very few substantive comments. As an initial study of a complex phenomenon it has many drawbacks but to the credit of the authors many of those are described in detail in section 4.

We thank Professor Roderick for the positive feedback on the paper and constructive comments to improve the paper. We reply in detail to the comments below whereby the reviewer comments are presented in blue and our reply is in black. The line numbers refer to the new version of the manuscript.

Methods:

- One aspect of the project design/methodology that was not commented upon is the inclusion of the Budyko process to separate P into Q and ET. I did not understand why this step was included since the climate model projections include ET and Q and you could use those directly. (See Roderick et al 2015; Milly & Dunne 2017). (Also see point 3 below.)

  The climate model projections indeed include ET and Q but retrieving these directly from the climate model projections would not allow us to show the exact effects of the tree cover change on the ET and Q fluxes. With the Budyko calculation approach we can first calculate the ET and Q fluxes with a 'basis' tree cover (used in scenario present climate and scenario climate change) and thereafter we calculate the ET and Q fluxes with a new tree cover (used in scenario climate change and tree cover change). The difference in ET and Q fluxes under climate change is then ascribed to the changing tree cover. With this approach we assume a consistent tree cover (per scenario) for the five CMIP6 climate models in this study and thus that we utilize the exact same tree cover change and the same ET methodology in our calculations for each of the five models and each of the scenarios.

- I think it might be useful to extend the discussion a little further to give full context. The idea of planting trees to their "potential" and then assessing the hydrologic changes is challenging but useful. Overall the global impacts of tree planting (or climate change) on runoff were small (Fig. 3) but could be important locally. Imagine we actually did plant all of those trees. The other impacts are on biodiversity (likely increase) but the big impact would be on agricultural production which would decrease (by a lot). Something should be said about this in the discussion.

  Our study is hypothetical, but planting all of those trees could indeed have important impacts on biodiversity and agricultural production. We propose the following paragraph addressing this subject in discussion section 4.1, starting at L415:

  '*It should be kept in mind that although our study is hypothetical, the realization of the potential tree cover in a future climate can have widespread consequences for biodiversity and agricultural production. For example, tree planting based on the future potential tree cover map could negatively impact biodiversity as the map permits afforestation in grassy biomes, for example in the Mississippi River basin. These regions can naturally support trees but host very different species and therefore afforestation would lead to large losses in biodiversity (Veldman et al., 2015). In addition, the future potential tree cover map shows a high potential for tree cover changes on agricultural (and urban) land (Section 2.2), while actual reforestation in these areas is likely limited to maintain agricultural production (Roebroek, 2023).*'

1. Line 16. "…. Water availability with more than 5 mm yr-1 ." I did not understand what the 5 mm yr-1 was referring to?

   This sentence refers to the information provided in Fig. 7 of the manuscript, whereby 'more than 5 mm yr-1' indicates the extent of water availability decrease as a result of climate change and tree cover change. In other words, this sentence explains that a part of the land surface (14 %) could locally experience relatively large decreases in water availability. To avoid confusion here, we suggest to change the sentence to (L15);

   *'However, for 14 % of the land surface, both tree cover change and climate change could decrease water availability by more than 5 mm yr-1.'*

2. Line 34. Typo. … increased by more

   Thank you, corrected.

3. Line 403-404. Yes, the Budyko models do not explicitly account for that but the climate models do (see Yang et al 2019, Hydrologic implications of vegetation response to elevated $CO_2$ in climate projections, Nature Climate Change). This begs the question of why use the Budyko models to split the P between ET and Q. You could use climate model output directly and avoid the Budyko step in the methodology.

   As mentioned above, we indeed use the Budyko models as a simplified approach which allows us to adjust the tree cover and thereafter assess the exact impact on the water fluxes as a result of changing the tree cover. Retrieving the water fluxes directly from the climate models for scenario 'climate change + tree cover' would require us to run those climate models with an altered tree cover.

References:

Roebroek, C. T.: Exploring the limits of forest carbon storage for climate change mitigation, Doctoral thesis, ETH Zurich, https://doi.org/10.3929/ethz-b-000655156, 2023.

Veldman, J. W., Overbeck, G. E., Negreiros, D., Mahy, G., Le Stradic, S., Fernandes, G. W., Durigan, G., Buisson, E., Putz, F. E., and Bond, W. J.: Where Tree Planting and Forest Expansion are Bad for Biodiversity and Ecosystem Services, BioScience, 65, 1011–1018, https://doi.org/10.1093/biosci/biv118, 2015.

**Reviewer 2**

This study investigates the hydrological impacts of large-scale tree cover changes and climate changes using the Budyko framework with data from CMIP6 models, tree cover datasets, and the UTrack moisture recycling dataset. The authors claim that this study provides a first estimate of the impacts of tree cover change under climate change on global hydrological fluxes over land. However, the novelty of the methods and conclusions is significantly overstated. Numerous studies have already addressed this topic, and the methodological flaws prevent me from recommending this paper for publication at its current stage.

We thank reviewer 2 for their feedback comments on the paper, and the suggestions to improve the manuscript. We reply in detail to the comments below whereby the reviewer comments are presented in blue and our reply is in black. The line numbers refer to the new version of the manuscript.

My major concerns are as follows:

1. The study estimates ET and Q using the Budyko equations, assigning fixed parameters for scenarios with and without tree cover. This approach predetermines the impact of tree cover change on the water balance, reducing the subsequent calculations to a numerical exercise with limited scientific value. Moreover, the implicit assumption behind this method—that changes in Budyko parameters are solely related to tree cover—is problematic. Climate factors, such as snow proportion and rainfall intensity, also influence Budyko parameters and affect the estimates of ET and Q. At a minimum, the authors should validate their approach by demonstrating its ability to estimate ET and Q using observational data.

   We agree that the Budyko parameters are also affected by other factors besides vegetation, such as soil water characteristics (Gunkel and Lange, 2017), or snowfall conditions (Berghuijs et al., 2014). Our assumption that the changes in Budyko parameters are solely related to tree cover is therefore indeed simplified, however, we disagree that this is problematic for our study. Several studies (e.g. Zhang et al., 2001) have shown that vegetation type explains a large part of the variability in rainfall-runoff ratios on a multi-annual scale, indicating that Budyko models can be applied to study the impact of vegetation on runoff. By using six different Budyko models, we provided insight into the uncertainty of the Budyko calculations, which is (among others) related to the fraction of variability in runoff that is not explained by vegetation.

   Furthermore, we do not validate the research approach in our manuscript since this methodology was already tested by Hoek van Dijke et al. (2022). In that study, the mean streamflow (Q) obtained with the Budyko calculations for a present climate was validated with observational data for 19 large river basins (see Fig. 1b in Hoek van Dijke et al. (2022)). We will refer to this methodology validation by Hoek van Dijke et al. (2022) in the revised version of the manuscript:

   L150: *'This approach was tested by Hoek van Dijke et al. (2022) whereby the mean streamflow (Q) obtained with Budyko model calculations for the present climate was validated with observational data for 19 large river basins.'*

   The study of Hoek van Dijke et al. (2022) applied Budyko calculations and the UTrack moisture tracking dataset to analyse the influence of large-scale potential tree cover change on hydrological fluxes. Their results generally align with the findings of Tuinenburg et al. (2022) who calculated the effect of potential tree cover change on present climate evaporation with a global hydrological model and the UTrack moisture tracking model. Hence, all of the above added to our confidence that we could utilize the Budyko models (and the UTrack datasets) to estimate the changes in hydrological fluxes following tree cover change.

2. The logic behind combining historical UTrack data with projected future precipitation to represent the scenario of 'Climate change with tree cover change with altered moisture recycling' is highly confusing. These datasets cover different time periods, and the authors themselves admit that "The UTrack dataset does not account for the feedbacks of altered tree cover." Additionally, forest-climate feedbacks are complex and can affect net radiation through changes in albedo, surface temperature, and other factors, thereby influencing PET. Why do the authors focus solely on precipitation changes without considering the potential impacts on the energy balance?

We understand that our use of the historical UTrack moisture tracking dataset for a future climate may seem confusing. However, as also mentioned in lines 194 - 203 of the manuscript, there is not yet a global moisture tracking dataset readily available for the future time period under climate change. Therefore, the UTrack moisture tracking dataset and the climate change scenarios cover different time periods as we are limited by the availability of future moisture tracking data. This shortcoming was clearly stated in Section 2.4 and the implications were discussed in Section 4.2 of the manuscript.

We agree with the reviewer that forest-climate feedbacks are complex and that our methodology does not account for feedbacks of tree cover change on atmospheric moisture recycling and the energy balance. Including these feedbacks in the context of this study would require running the five CMIP6 models - with a coupled biosphere and atmosphere - both with and without large-scale tree cover change, which could introduce other uncertainties (discussed in lines 469 - 475 of the manuscript). We already briefly address that our approach omits the feedbacks of a changing tree cover on the energy balance through e.g. surface temperatures and albedo, and we will extend this section of the discussion on the implications of leaving out the potential impacts on the energy balance. Below you can find the suggested additional text for the discussion section, starting from L437 onwards:

*There are no moisture tracking datasets available for an Earth with high tree cover and / or climate change, which is why the current UTrack moisture recycling dataset is used. Below, we describe the feedbacks that were omitted, and the effect that it has on our results.*

*This research does not account for the feedbacks of tree cover change on moisture recycling via changes in e.g. the surface albedo, cloud cover, atmospheric carbon dioxide concentrations, surface temperatures, length of moisture transport pathways, and global circulation. These feedbacks can impact the energy balance, e.g. increased tree cover can lower the surface albedo and subsequently enhance the surface temperature, and impact global atmospheric circulation and moisture transport (Portmann et al., 2022). However, these feedbacks are complex and can be contrasting, depending on the location of the land use change.*

3. There is a lack of coherence between the simulation results and tree cover datasets. The authors use the global tree cover dataset provided by Hansen et al. (2013) as the tree cover data for the 'present' scenario, but the climate models they use for the 'present' scenario do not incorporate this tree cover dataset. The same issue arises for future simulations with potential tree cover changes. In conjunction with my first point, it seems that the study's methodology is more of a numerical combination of different datasets rather than a robust scientific analysis. This raises doubts about the reliability of the results.

In this research we use a consistent tree cover across all CMIP6 models for each research scenario, instead of varying the tree cover for each model. Every CMIP6 model has its own land cover map, and we are aware that the tree covers in our approach deviate from those in the CMIP6 models. We chose this method for two different reasons;

1. The Budyko model calculations are performed for each of the research scenarios and CMIP6 models. By using a consistent tree cover dataset across the five CMIP6 models in the Budyko calculations, we exclude the impacts of model-related tree cover on the calculated hydrological fluxes and thereby minimize flux differences between the models. Hence, we aim that the variations in calculated hydrological fluxes across the CMIP6 models are solely related to differences in model climate conditions.

2. For each of the CMIP6 models, the future potential tree cover by Roebroek (2023) is based upon the tree cover by Hansen et al. (2013) instead of the tree cover corresponding to the CMIP6 model. Hence, in our research we use the tree cover by Hansen et al. (2013) as the 'base' present tree cover to ensure consistency between the present tree cover and the future potential tree cover.

For the future potential tree cover we underline that the tree cover under the SSP3-7.0 pathway in CMIP6 models deviates from the future potential tree cover used in our approach. Therefore, the feedback loop between potential tree cover change and climate conditions (a changing tree cover would alter the climate conditions which in turn would change the tree cover, and so on) is not accounted for in this research. We discuss the limitations regarding the usage of the future potential tree cover map and the exclusion of feedback loops corresponding to tree cover change starting at L390 of the manuscript.

Finally, we do not agree with the reviewer that our study is a 'numerical combination of different datasets rather than a robust scientific analysis'. Numerical combinations are the basis for many scientifically robust studies. Our data-driven approach has its shortcomings, as also clearly discussed in the manuscript, but it also has its strengths and faces different sources of uncertainty than earth system model studies. For example, the Budyko models allow us to account for the feedbacks on soils, rooting systems, and litter layer following (re)forestation, whereas these processes are omitted in most earth system models. Furthermore, a recent model-based study by King et al. (2024) analyses the hydrological impacts of (re)forestation in a future climate using a single climate model, therefore not considering the biases present in global climate models. Since our study adopts a multi-model approach by including data from five CMIP6 models, our methodology allows us to account for biases in e.g., precipitation which can differ substantially per model (Fig. 3a of the manuscript). It should also be noted that the predicted effects of climate change on vegetation are highly uncertain in earth system models (Terrer et al., 2019; Roebroek et al., 2024).

Overall, we believe that our study complements earth system models studies, which encounter different sources of uncertainties, and thereby contributes to this topic from a different perspective. We clearly acknowledge the constraints of this study and present our results as a first estimate of the hydrological impacts of tree cover change in a future climate.

4. The analysis is overly simplistic. While the paper devotes significant space to introducing tree cover change, it notes that tree cover change "is not used for calculations and is solely used to visualize the differences in tree cover between the present climate and CC, and the CC+TCC scenarios."

We agree with the reviewer that this sentence was a bit unclear and could be interpreted incorrectly, therefore we removed it from the revised version of the manuscript to avoid further confusion.

The final result only reiterates that while climate change increases global runoff, changing tree cover reverses this effect, leading to a limited net impact on runoff compared to the present climate and current tree cover. This conclusion has been well documented in the literature, and the paper does not offer any additional insights.

We disagree with the reviewer that our conclusion has been well documented in previous literature. There have indeed been other publications which addressed this topic, however, most of these studies were either on a different spatial scale (e.g., Buechel et al. (2024) focused on the UK), focused on one earth system model (King et al., 2024), focused on a different climate pathway (e.g., Buechel et al. 2024; King et al., 2024), or did not consider climate change (Hoek van Dijke et al., 2022). Our study provides a global overview of how large-scale tree cover change under climate pathway SSP3-7.0 can amplify, mitigate, or reverse negative effects of climate change on runoff. Hence, while the globally averaged impacts of climate and tree cover changes indeed result in a limited net change in runoff, we also show that their effects on water availability can differ substantially on a regional scale.

Finally, as mentioned before, we consider it an asset if this complex topic is studied from various research perspectives, by using different methodologies and recording the corresponding uncertainties.

5. On a positive note, the authors have acknowledged many of these issues in their discussion. However, merely acknowledging these limitations is not enough to mitigate their impact on the reliability and scientific integrity of the paper. A more rigorous scientific approach is needed to explore this important and interesting topic.

There are indeed shortcomings in our study and given these constraints we present our results as a first estimate of the hydrological effects due to climate change and tree cover change. By underlining the shortcomings of our research methodology we provide future studies with potential next steps and guidelines for further improvement and validation of our findings. As mentioned before, we believe that each methodology has specific shortcomings and advantages and by following a diversity of approaches in different studies, we can generate a clear overview of this challenging topic.

References:

Berghuijs, W., Woods, R. & Hrachowitz, M.: A precipitation shift from snow towards rain leads to a decrease in streamflow, Nature Clim Change 4, 583–586, https://doi.org/10.1038/nclimate2246, 2014.

Buechel, M., Berthou, S., Slater, L., Keat, W., Lewis, H., and Dadson, S.: Hydrometeorological response to afforestation in the UK: findings from a kilometer-scale climate model, Environmental Research Letters, 19, https://doi.org/10.1088/1748-9326/ad4bf6, 2024.

Gunkel, A., & Lange, J.: Water scarcity, data scarcity and the Budyko curve—An application in the Lower Jordan River Basin, Journal of Hydrology: Regional Studies, 12, 136-149. https://doi.org/10.1016/j.ejrh.2017.04.004, 2017.

Hansen, M. C., Potapov, P. V., Moore, R., Hancher, M., Turubanova, S. A., Tyukavina, A., Thau, D., Stehman, S. V., Goetz, S. J., Loveland, 580 T. R., Kommareddy, A., Egorov, A., Chini, L., Justice, C. O., and Townshend, J. R. G.: High-Resolution Global Maps of 21st-Century Forest Cover Change, Science, 342, 850–853, https://doi.org/10.1126/science.1244693, 2013.

Hoek van Dijke, A. J., Herold, M., Mallick, K., Benedict, I., Machwitz, M., Schlerf, M., Pranindita, A., Theeuwen, J. J. E., Bastin, J.-F., and Teuling, A. J.: Shifts in regional water availability due to global tree restoration, Nature Geoscience, 15, 363–368, https://doi.org/10.1038/s41561-022-00935-0, 2022.

King, J. A., Weber, J., Lawrence, P., Roe, S., Swann, A. L. S., and Val Martin, M.: Global and regional hydrological impacts of global forest expansion, Biogeosciences, 21, 3883–3902, https://doi.org/10.5194/bg-21-3883-2024, 2024.605

Roebroek, C. T.: Exploring the limits of forest carbon storage for climate change mitigation, Doctoral thesis, ETH Zurich, https://doi.org/10.3929/ethz-b-000655156, 2023.

Roebroek, C. T. J., Caporaso, L., Alkama, R., Duveiller, G., Davin, E. L., Seneviratne, S. I., & Cescatti, A.: Climate policies for carbon neutrality should not rely on the uncertain increase of carbon stocks in existing forests, Environmental Research Letters, 19(4), https://doi.org/10.1088/1748-9326/ad34e8, 2024.

Terrer, C., Jackson, R. B., Prentice, I. C., Keenan, T. F., Kaiser, C., Vicca, S., Fisher, J. B., Reich, P. B., Stocker, B. D., Hungate, B. A., Peñuelas, J., McCallum, I., Soudzilovskaia, N. A., Cernusak, L. A., Talhelm, A. F., Van Sundert, K., Piao, S., Newton, P. C. D., Hovenden, M. J., Blumenthal, D. M., Liu, Y. Y., Müller, C., Winter, K., Field, C. B., Viechtbauer, W., Van Lissa, C. J., Hoosbeek, M. R., Watanabe, M., Koike, T., Leshyk, V. O., Polley, H. W., & Franklin, O.: Nitrogen and phosphorus constrain the $CO_2$ fertilization of global plant biomass, Nature Climate Change, 9(9), 684-689. https://doi.org/10.1038/s41558-019-0545-2, 2019.

Tuinenburg, O. A., Bosmans, J. H. C., and Staal, A.: The global potential of forest restoration for drought mitigation, Environmental Research 685 Letters, 17, 1–8, https://doi.org/10.1088/1748-9326/ac55b8, 2022.

Zhang, L., Dawes, W. R., and Walker, G. R.: Response of mean annual evapotranspiration to vegetation changes at catchment scale, Water 710 Resources Research, 37, 701–708, https://doi.org/10.1029/2000wr900325, 2001.

**Reviewer 3**

The study by Engel et al. presents a very interesting approach to examining the hydrological impacts of large-scale tree cover change under future climate scenarios. The interdisciplinary method you employ, combining data from multiple CMIP6 climate models, Budyko models, and the UTrack dataset, provides a good initial estimate of how climate change and tree cover shifts may influence water availability.

The author's acknowledgment of the limitations, particularly the inability of the UTrack dataset to capture energy balance changes in both current climate (CC) and future tree cover change (TCC) scenarios, is well-placed. I appreciate that you have addressed these important limitations in detail within the methodology and discussion sections, providing clarity on the scope of your findings.

Despite these constraints, the manuscript still offers valuable insights into the potential hydrological consequences of tree cover change at a global and regional scale. The authors highlight the complex interplay between climate-driven and vegetation-driven effects on runoff. Future studies that could take a more complex approach and employ fully coupled models and could build on your findings to provide an even more comprehensive understanding.

We thank reviewer 3 for their positive and constructive feedback comments on the paper. We are happy to read that the reviewer appreciates our discussions of the uncertainties and limitations of the study. We reply in detail to the comments below whereby the reviewer comments are presented in blue and our reply is in black. The line numbers refer to the new version of the manuscript.

- Visualization of Table 1: Consider redrawing Table 1 as a flow chart to clarify the workflow. This could improve the reader's understanding of your methodology at a glance.

  We thank the reviewer for raising this point and agree that a simplified flow chart would provide a clear and more accessible overview of our methodology. Therefore, we will add a simplified flow chart (indicating our main research steps) to the methodology section whilst also retaining Table 1 in this section to visualize the detailed layout of the methodology. The proposed flow chart is shown below;

[Figure]

*Figure 2. Simplified overview of the research methodology. This figure shows the research scenarios: 1) scenario present climate; 2) scenario climate change (CC); 3) scenario climate change with tree cover change with changed moisture recycling (CC + TCC); as well as the*

*intermediate research steps. A detailed overview of the input and output data for each research step can be found in Table 1.*

- Clarity in Methodology: The beginning of the methods section could be more accessible. I suggest explaining the necessity of including multiple Budyko models for the uncertainty estimate earlier in the section to guide readers through your approach. Streamlining Table 1, possibly by replacing it with a simplified flow chart, and moving the detailed Table 1 to the appendix could help improve clarity.

We thank the reviewer for raising this point and as also mentioned above we will insert a simplified flow chart for additional clarity. Furthermore, we suggest the following changes in at the start of the methodology section, whereby Fig. 2 would refer to the flowchart:

L89 - 98: *'These three are: 1) scenario present climate, 2) scenario climate change (CC), and 3) scenario climate change with tree cover change and moisture recycling change (CC + TCC) (Fig. 2, Table 1). We calculate the effects of climate change as the difference between scenario present climate and scenario CC, and calculate the effects of tree cover change as the difference between scenario CC and scenario CC + TCC (Fig. 2). For each research scenario, we use P and potential evapotranspiration (PET) datasets from five CMIP6 climate models (Sect. 2.1), along with a tree cover dataset (Sect. 2.2), as input for the Budyko model calculations (Sect. 2.3) to generate ET and Q fluxes. Furthermore, scenario CC + TCC includes the (non-)local indirect effects of changes in ET and P by accounting for an altered moisture recycling, obtained with the moisture tracking dataset UTrack (Sect. 2.4). We therefore build on the general research methodology from Hoek van Dijke et al. (2022), extending their approach to assess the hydrological effects of climate and tree cover change under future climate conditions.'*

We also agree with the reviewer that we should more clearly state the necessity of including multiple Budyko models, however, we prefer addressing this in the section dedicated to the Budyko models (Section 2.3). We suggest the following explanation (L158):

*'Each of these Budyko models was calibrated with lysimeter or streamflow data originating from different river basins, thereby representing different vegetation types (e.g. plantation or natural vegetation, or deciduous or evergreen forest) and climate conditions. Therefore, we include multiple Budyko models for our global scale calculations to represent the spread between different models and minimize potential biases related to e.g. climate conditions.'*

- Use of the UTrack Dataset: The decision to use the UTrack dataset at a 1° resolution, when it is available at 0.5°, warrants an explanation.

The datasets of the CMIP6 models that fulfilled our predetermined conditions (mentioned in lines 113 - 117 of the manuscript) were only available at a relatively coarse spatial resolution (see also Table 2 in the manuscript). Therefore, we decided to use the UTrack dataset at the coarser spatial resolution of 1° by 1° instead of remapping the climate datasets to a higher spatial resolution. Additionally, we perform all calculations on this 1° by 1° spatial resolution for consistency. We will suggest the following sentence for the revised version of the manuscript to illustrate our motivation for using the 1° by 1° UTrack dataset, starting at L181:

*'The UTrack datasets are available at 0.5° and 1° spatial resolutions, however, we utilize the coarser spatial resolution of 1° by 1° as the datasets of the selected CMIP6 models in our study are only available at relatively coarse spatial resolutions (Table 2).'*

Additionally, a recent preprint (https://www.researchsquare.com/article/rs-4177311/v2) has highlighted a potential issue with the water balance in the dataset, which should be acknowledged, particularly since it is being utilized for water balance estimations. Ensuring

that the global water balance checks out would strengthen the validity of your analysis.
Thank you for notifying us of this paper. We will refer to the paper in section '2.4 Utrack moisture recycling dataset' (L177). During our analyses we indeed checked for closure in the water balance, whereby we found an almost negligible difference ($\times 10^{-14}$) between the global change in ET and global change in P, which we related to rounding effects.

Furthermore, here are two papers that may be relevant, as they utilize the UTrack dataset for basin-level estimations and also account for the impact of land use changes. These references could provide additional context and support for your analysis (https://www.nature.com/articles/s44221-024-00291-w, https://doi.org/10.1029/2023EF003837).
Thank you for bringing these papers to our attention. The paper by Fahrländer et al (2024) fits well in our introduction, and we will cite it in L28.

References:

Fahrländer, S.F., Wang-Erlandsson, L, Pranindita, A., and Jaramillo, F.: Hydroclimatic Vulnerability of Wetlands to Upwind Land Use Changes, Earth's Future, 12, 3, e2023EF003837, https://doi.org/10.1029/2023EF003837, 2024